# The Working Environment in Primary Healthcare Outpatient Facilities: Assessment of Physical Factors and Health Professionals’ Perceptions of Working Environment Conditions

**DOI:** 10.3390/ijerph21070847

**Published:** 2024-06-28

**Authors:** Marta Regina Cezar-Vaz, Clarice Alves Bonow, Joana Cezar Vaz, Carlos Henrique Cardona Nery, Mara Regina Santos da Silva, Daniela Menezes Galvão, Aline Soares Alves, Flávia Santana Freitas Sousa, Joice Simionato Vettorello, Jociel Lima de Souza, Joaquim Vaz

**Affiliations:** 1School of Nursing, Federal University of Rio Grande, Rio Grande 96203-900, Brazil; denfmara@furg.br (M.R.S.d.S.); daniela.galvao@furg.br (D.M.G.); alinesalves@furg.br (A.S.A.); 2Faculty of Nursing, Federal University of Pelotas, Pelotas 96010-610, Brazil; clarice.bonow@ufpel.edu.br; 3Financial Planning Department, Vibra Energia Company, Rio de Janeiro 20211-140, Brazil; joanavaz@vibraenergia.com.br; 4Institute of Human and Information Sciences—ICHI, Federal University of Rio Grande, Santa Vitória do 11 Palmar Campus, Santa Vitória do Palmar 96230-000, Brazil; carloscardona@furg.br; 5University Hospital Professor Edgar Santos, Federal University of Bahia (Hupes-UFBA), Salvador 40110-060, Brazil; sousa.flavia@ebserh.gov.br; 6University Hospital Doctor Miguel Riet Corrêa Junior, Federal University of Rio Grande (HU-FURG), Rio Grande 96200-190, Brazil; joice.vettorello@ebserh.gov.br; 7Municipal Department of Administrative Management and Bids (SMGAL), Municipal Government, Rio Grande 96200-015, Brazil; jociel.souza@riogrande.rs.gov.br; 8School of Engineering, Federal University of Rio Grande-Carreiros Campus, Rio Grande 96203-900, Brazil; joaquimvaz@furg.br

**Keywords:** physical factors, temperature, air humidity, noise, illuminance, occupational health, risk perception, security and healthy environmental, primary healthcare

## Abstract

The objectives of this study were to assess the adequacy of physical parameters/factors (temperature, relative humidity, noise, and illuminance levels) of the work environment in PHC facilities, to evaluate the association between the adequacy of these measured physical parameters and the physical characteristics of the PHC facilities and their surroundings and to assess the association between health professionals’ perceptions about exposure to physical risks in the PHC work environment and the adequacy of physical parameters measured in the same facilities. The study monitored 23 PHC facilities in southern Brazil and interviewed 210 health professionals. Data analysis involved Pearson’s chi-square, Fisher’s exact test, Spearman’s correlation, and multivariate linear regression analysis was used to control for confounding factors. The significance level was set at 5% (*p* ≤ 0.05). The combination of temperature and relative humidity presented thermal comfort levels outside the adopted criteria for adequacy in consultation (outdoor relative humidity, *p* = 0.013) and procedure rooms (front door open, *p* = 0.034). Inadequate sound comfort (noise) levels in the morning shift were found in the vaccination (front door open, *p* = 0.021) and consultation rooms (movement of people, *p* = 0.016). In PHC facilities where reception rooms had insufficient lighting, internal curtains were opened less frequently (*p* = 0.047). The analysis of health professionals’ perceptions of physical factors demonstrated that physicians more frequently perceive the physical risk of temperature and humidity (*p* = 0.044). The higher the number of nurses (*p* = 0.004) and oral health technicians in the PHC facilities (*p* = 0.031), the greater the general percentage of adequacy of monitored physical parameters. It was also confirmed that the higher the perception of moderate or severe physical risk among health professionals, the lower the general percentage of the adequacy of the physical parameters of the work environment of the PHC facilities evaluated (r_s_ = −0.450, *p* = 0.031). This study’s evidence contributes to a better understanding of physical conditions and future occupational interventions to ensure the comfort, safety, and well-being of PHC workers.

## 1. Introduction

This manuscript describes a study addressing working conditions, specifically focusing on physical environmental factors measured on-site in the context of primary healthcare facilities (UBS in Portuguese) integrating the primary healthcare (PHC) network in a city in southern Brazil. Physical factors or parameters of working environments, such as temperature and relative humidity, noise, and illuminance, are important factors influencing the working conditions of healthcare facilities [1,2,3]. However, both objective and subjective measures can be adopted to describe the conditions of working environments. Subjective measures include the workers’ perceptions of their exposure to physical factors and how these factors influence daily tasks and, consequently, their well-being [4] and productivity levels [5]. The scientific literature has already associated physical factors (e.g., temperature and relative humidity, noise, and illuminance levels) and the workers’ perceptions of how intensively they are exposed to these factors to healthy (or unhealthy) working conditions [6]. Nevertheless, both aspects (objective measures and subjective perceptions) lack a more profound understanding in the context of PHC facilities [7,8], which is the focus of this study.

Studies addressing the physical factors of working environments include different parameters. Knowledge regarding such factors and the methods adopted in studies indicate the issues that must be considered when outlining policies designed to regulate working environments to ensure decent and healthy working conditions [9,10]. Although evidence indicates that physical factors positively/negatively influence the working conditions in healthcare facilities, workers’ perceptions do not always clearly express such influence. In other words, the workers’ perceptions regarding exposure to physical risks may not correspond to the objective measures of physical factors encountered in a working environment, considering the required parameters to ensure healthy workplaces with adequate work conditions, e.g., adequate temperature and relative humidity, noise, and illuminance levels, which are the factors addressed here.

Naturally, the physical parameters addressed in this study comprise a small portion of the factors that ensure adequate (or inadequate) working environments [10,11]. Therefore, one must use caution when referring to these factors to express physical comfort in working environments [6,12]. Nevertheless, the concept of physical comfort in working environments was consciously adopted in this study, considering that this is a pilot study with limitations and contributions. Considering technical and scientific aspects, detailed and systematic procedures were adopted to support the interests of decision-makers and managers, including those in public organizations, at the regional, national, and international levels. Hence, similarities and differences concerning physical constraints were identified in PHC facilities [13,14].

Primary healthcare facilities materialize the physical infrastructure, equipment, human resources, materials, and inputs necessary to provide PHC services. They are strategically designed to be located in the same physical environment as the communities they serve [13,14]. In different countries, these physical facilities and organizations are the population’s first access to the public health system [15], in this case, the Brazilian public health system [16].

In this context, valuing the relationship between the physical conditions of working environments and the workers’ perceptions regarding physical risks (which may hinder or facilitate the work of PHC multidisciplinary teams) contributes to the proactive management of negative and positive occupational conditions, ensuring occupational safety and health, and health surveillance. The work environment of PHC facilities refers to the healthcare model currently in force in the Brazilian Unified Health System (SUS) [17], including actions focused on well-being promotion and disease prevention, as well as heeding potential injuries and referring the most severe cases to facilities providing more complex services with technological support [13]. PHC is where the healthcare network is organized, and comprehensive healthcare is coordinated and delivered to the population [13,15]. In terms of occupational health, a PHC facility incorporates the physical conditions of the working environment that determine the health–disease continuum [18,19]. Additionally, the variables addressed here are based on the concepts provided by the Brazilian Ministry of Labor and Social Security as expressed in Regulatory Standards (NR), the Brazilian Regulatory Standards (NBR), and the Brazilian Association of Technical Standards (ABNT), which are based on international guidelines, such as the International Organization for Standardization (ISO) [12].

Hence, the physical conditions of working environments are included in the broader and universal concepts of healthy, decent, and safe work conditions provided by the International Labor Organization—ILO [20]. The relationship between labor and health is reciprocal, i.e., healthy, safe, and decent working conditions protect and improve the health of workers, as well as the health and well-being of their families and the community [21].

Therefore, this study focuses on a small but relevant portion of this broad and universal concept, which determines adequate (inadequate) working conditions (healthy, safe, and decent work) among healthcare workers [20]. Thus, this study addresses the physical factors of the work environment in primary healthcare facilities and how healthcare workers perceive the risks associated with these physical factors. Considering that these factors can represent risks for both workers and patients, the objectives of this study were (a) to assess the adequacy of physical parameters/factors (temperature, relative humidity, noise, and illuminance levels) of the work environment in PHC facilities; (b) to evaluate the association between the adequacy of these measured physical parameters and the physical characteristics of the PHC facilities and their surroundings; (c) to assess the association between health professionals’ perceptions about exposure to physical risks in the PHC work environment and the adequacy of physical parameters measured in the same facilities. In this way, it was possible to question whether the PHC facilities with the highest percentage of adequacy regarding the four criteria (for parameters) were also those with the lowest percentage of health professionals who perceived physical risk in the work environments of the PHC facilities.

According to the proposed objectives, the following hypotheses were extracted: (a) the prevalence of adequacy of physical parameters in the work environment in the two shifts of the four rooms in the 23 PHC facilities evaluated is low; (b) the physical characteristics of each installation and its surroundings are associated with the adequacy of the physical parameters of the rooms, such as temperature and humidity; (c) there are associations between the perceptions of health professionals working in the PHC facility and the adequacy of physical parameters measured in the same facilities.

## 2. Materials and Methods

### 2.1. Origin of the Current Study

This study integrated the macro project titled The Socio-Environmental Dimension of the Health of PHC Workers in the South of Brazil (“Dimensão Socioambiental da Saúde dos Trabalhadores da APS do Sul do Brasil”). The National Council for Scientific and Technological Development (CNPq) provided financial support. The research team consists of experienced researchers in the areas of public health and engineering and health professionals in training in the doctoral program in health sciences, permanent members or guests of the Laboratory for the Study of Socio-Environmental Processes and Public Health Production (LAMSA) from the Federal University of Rio Grande, and researchers from other partner universities. The broader study was conducted in two municipalities in the extreme South of Rio Grande do Sul, Brazil. When structuring and operationalizing this project, Municipality 1 had medium-sized characteristics and 31 PHC outpatient units, while Municipality 2 was small-sized and had 10 PHC outpatient facilities/units [22]. The Epi Info^®^ StatCalc tool (version 7.2, CDC, Atlanta, GA, USA) was used to estimate the sample size in the broader study. During the study period, 548 typical PHC workers were considered following the Brazilian Government’s National Health Policy [22], including nurses, doctors, dentists, nursing technicians/aides, community health agents, and oral health technicians/aides. A margin of error of 5%, a confidence level of 95%, and losses of 5% were established. The professionals were selected using non-probability sampling. At least 232 professionals from the covered area should be included in the consecutive intentional sample [23]. The inclusion criteria were working in a PHC service for at least six months, while the exclusion criteria were absent during data collection (from January to March 2020). In the broader study, 342 health professionals were interviewed in two municipalities. Additional information about the participants in the broader study can be found in the previous study [24].

### 2.2. Current Study Design, Sample, and Setting

This current study is composed of two integrated parts: the first is a cross-sectional study that measures the physical parameters/factors (temperature, relative humidity, noise, and illuminance levels) of the working environment of PHC facilities, and the second is an ecological survey that analyzes the perception of health professionals regarding exposure to physical risk factors of the work environment in the same PHC facilities studied.

Municipality 1 was the only place where this study was conducted. The reason for selecting Municipality 1 is its role as one of the region’s representative municipalities concerning the organization of the PHC network [22]. It has the most significant number of PHC facilities, being the host city of the Federal University of Rio Grande, which maintains collaborative training and research work with the city’s Health Department; as well as for the logistics that involved the entire process for collecting data regarding the monitoring of the physical parameters/factors of the PHC facilities. 

The cross-sectional study on the physical parameters/factors (temperature, relative humidity, noise, and illuminance levels) of the PHC work environment was conducted in 23 PHC facilities in Municipality 1 from May to June 2022. It should be noted that these facilities were selected because they are the same PHC facilities where interviews with health professionals were conducted. Given the recommendations for the physical structure of the PHC facilities [13] to ensure the complex functioning of services, the fieldwork was restricted to four internal areas that are common in the organizational arrangement of the PHC network and represent the purposes of use of the environments composing a PHC facility: the reception room (RR), the vaccination room (VR), the consultation room (CR), and the technical procedures room (PR). Hence, the physical factors concerning the temperature, relative humidity, noise, lighting/illuminance levels, and the rooms’ physical conditions were assessed in all of the PHC facilities.

The ecological study of professionals’ perception of exposure to physical risk factors in the work environment of PHC facilities was carried out using the consecutive intentional sampling [23]. The sample consisted of all 210 health professionals who were interviewed at 23 PHC facilities in Municipality 1 from January to March 2020. The participants were distributed as follows: 25 nurses, 25 physicians, 43 nursing technicians, 100 community health agents, 7 dentists, 8 oral health technicians/assistants, and 2 others. Data from participants included in the sample were extracted from the broader study database indicated previously. These data were organized with data on the physical parameters of the PHC facilities’ work environments for posterior analyses.

### 2.3. Measure

#### 2.3.1. Physical Parameters/Factors

An instrument was created and tested to measure physical factors such as temperature, relative humidity, noise levels, and illuminance in primary healthcare (PHC) facilities. The instrument was developed with the support of three experts in the field of human comfort in a built environment. Two types of equipment were used for the measurements: a “Multiparameter Environment Meter (5 in 1) KR500” by AKROM and a “Laser distance meter GM 500–Trena” by BOCRH, which were used to mark the distances between the measurement points.

A pilot study was conducted in two PHC units to evaluate the data collection instrument for physical parameters. Data from these two units were not included in the analysis, as they allowed the need for adjustments to the instrument to be identified. In another PHC unit, it was impossible to complete two fieldwork shifts, given the number of patients treated suspected of having COVID-19. Therefore, as previously mentioned, the data collected for analysis correspond to 23 PHC facilities, consisting of variables relating to physical parameters (temperature, relative humidity, noise, and illuminance levels), installation’s physical (structural) characteristics, surroundings, and geographic location.

The field team followed a predetermined schedule established during a meeting with the city’s PHC manager. Each PHC facility was monitored for one day, and the data were collected at two different times: in the morning (from 9 a.m. to 12 p.m.) and afternoon (from 1 p.m. to 5 p.m.). The data were collected from four representative indoor environments in each PHC facility, specifically the reception room (RR), vaccination room (VR), consultation room (CR), and technical procedures room (PR). Additionally, the data were collected from the outdoor environments closest to each indoor environment. The data were collected during the opening hours of the PHC facilities when the indoor environments were in use. The professionals at the PHC facilities kindly allowed the team to collect and record data regarding the conditions of use of the rooms. This included their identification, dimensions, solar orientation, windows and doors, as well as the measurement of the physical factors of interest, such as indoor and outdoor ambient temperature and relative humidity, indoor and outdoor noise levels, and indoor illuminance.

The physical (structural) characteristics of each facility and their surroundings and geographic location were classified as garden, trees, paved, lightly built, densely built, rural, or open field. The dimensions of the rooms were recorded, including their length, width, and height, and their orientation was classified as north, south, east, west, northeast, southeast, northwest, or southwest. Finally, the rooms’ structure was classified as masonry walls, lightweight partition walls, concrete slab, and lightweight ceiling panels. The physical factors monitored indoors (rooms) and outdoors were measured at specific points or locations. This detail can be seen in Appendix A.

The temperature and relative humidity levels were measured at the center of the rooms in both work shifts (Appendix A) and in the respective outdoor sites protected from solar radiation (Appendix A) at a height of 1.5 m. Yes, No, or Does not apply were the options adopted at each measurement to classify the room’s conditions of use considering the following items: front door open, outside window open, air-conditioned, fan on, refrigerator on, and other equipment on. The outdoor items were classified as sunny, partly cloudy, cloudy, or rainy.

Noise levels were also collected at the center of the room (Appendix A) and in a protected outdoor site (Appendix A) at a height of 1.5 m. Considering the equipment’s sensitivity, two consecutive measurements were taken at each site to prevent reading errors. The conditions of use at each indoor measurement were classified by answering Yes, No, or Does not apply to the following items: open door, outside window open, air-conditioned, fan on, refrigerator on, other equipment on, and outdoor noise. The items for the outdoor site included movement of people, people talking, motorcycle traffic, light vehicle traffic, heavy vehicle traffic, school noise, noise from businesses or workshops, rain, wind, or other types of noise.

Next, illuminance levels were taken at four symmetrical bilateral points at the working plane height to assess variations in the amount of light (Appendix A). The rooms’ conditions of use were classified at each measurement by answering Yes, No, or Does not apply to the following items: front door open, outside window open, curtains open, ceiling light on, and others. Outdoor lighting was not monitored.

#### 2.3.2. Health Professional’s Perception of the Physical Factors

A structured questionnaire was used to gather socio-demographic information, including age, self-reported race, marital status, education, and the number of children. It also collected information about the participant’s primary health care (PHC) occupation, such as type of profession, place of work, whether the participant had a second job, years of professional experience, years working in a PHC service, weekly working hours, PHC unit’s working hours, and monthly income [24,25]. The questions regarding professionals’ risk perception regarding physical risk factors used in this study were those related to the physical factors monitored, such as temperature, relative humidity, noise, and illuminance levels of the PHC work environment. It should be noted that the research group (LAMSA) used this structured questionnaire in other studies and found it suitable for the macro project that gave rise to the analyses carried out here.

During meetings with LAMSA members, the structured form’s variables, including socio-demographic, occupational, and risk perception variables, were tested and adjusted. Before data collection, a pilot study was conducted on ten individuals from different professions in the health field. The main goal of this pilot study was to determine the effectiveness and comprehension of the data collection instrument by evaluating the questions and improving the interviewing skills of field researchers.

During the months of January to March 2020, health professionals were invited and interviewed. At that moment, the COVID-19 pandemic had not yet caused the university to suspend its teaching and research activities. The team responsible for fieldwork underwent full training to collect data across the larger project. As part of the training program, the researchers covered various topics, including the methodology of data collection, work schedules, and how to establish relationships with professionals in primary healthcare facilities. The researchers also conducted face-to-face meetings with experts in human comfort in workplaces, occupational health, and work organizations in PHC to help everyone understand and use the data collection instruments and equipment effectively. To facilitate communication and collaboration, the researchers created a WhatsApp group for all those involved in the study. Through this group, the researchers aimed to share data collection and equipment usage knowledge and work together to achieve the goals.

Two or three researchers always worked together to ensure safety and speed up the selection process. The individual interviews with the PHC workers lasted an average of 58 min. The study’s objectives were explained to all participants, and they signed two copies of informed consent forms before joining. It should be noted that the methodological elements and procedures in this section were described in our previous studies with the same population and other outcomes [24,25].

### 2.4. Dados Analysis

(a)Physical Parameters/Factors

The suitability of the conditions of the PHC working environments was verified by comparing the measurements of the factors obtained in each room in two work shifts with the parameters recommended by normative standards. Regulatory Standard 17 (NR 17) is provided by the Brazilian Ministry of Labor and Social Security and deals with ergonomics [26]. Hence, it establishes guidelines and requirements that enable adapting and assessing working conditions to ensure worker comfort, safety, health, and efficient performance [10,11,12,26]. Other guidelines are provided by the Brazilian Ministry of Labor and Social Security, expressed in Regulatory Standards (NR) and Brazilian Standards (NBR) of the Brazilian Association of Technical Standards (ABNT), which are based on international guidelines, such as International Organization for Standardization (ISO) [12].

The concept of room adequacy was applied to each physical factor analyzed in this study (i.e., temperature, relative humidity, noise, and illuminance levels). Hence, when comparable parameters were found in both shifts, the room was considered adequate, partially adequate if acceptable parameters were found in just one shift, and inadequate otherwise.

#### 2.4.1. Temperature and Relative Humidity

The researchers collected 184 pairs of indoor measurements (temperature and relative humidity) in the four reference environments (rooms) in two shifts (morning and afternoon) in each of the 23 PHC facilities. Another 184 pairs of measurements (temperature and relative humidity) were collected in the corresponding outdoor sites. According to standard NR 17, temperature and humidity must be controlled in work environments to ensure thermal comfort, which, in air-conditioned environments, is obtained between 18 °C and 25 °C [26]. Hence, the criterion adopted to rate the suitability of the rooms in the PHC facilities was a temperature between 18 °C and 25 °C to ensure thermal comfort under the NR 17 [26], assuming relative air humidity between 40% and 60%, parameters currently considered ideal for human health and comfort [27,28,29].

#### 2.4.2. Noise

A total of 368 noise levels were collected indoors. These measurements correspond to two readings in four reference rooms in the morning and afternoon in the 23 PHC facilities. Another 368 noise levels were measured in the corresponding outdoor sites, as described in the method section. Two readings were taken in each shift to check the measurements. Later, in the analysis, the researchers decided to consider the average of the two readings. According to the NR 17 standard, comfortable background noise pressure levels are within the parameters established by official technical standards for indoor environments based on their purpose of use. For the remaining cases, acceptable background noise levels for acoustic comfort are up to 65 dB(A) [26]. This study compared the readings with the parameters recommended by the NBR 10.152 standard [30]. The following maximum acoustic levels were considered adequate according to the environments’ purpose of use: reception room—50 dB(A); vaccination and procedure rooms—45 dB(A); and consultation room—40 dB(A). Hence, the researchers compared the average noise levels on each measurement (morning and afternoon) taken in all the rooms and facilities to check whether the measurements were below or above the recommended maximum level [30,31].

#### 2.4.3. Illuminance

The recommended minimum illuminance for the environments addressed in this study was established based on an assessment of similar purposes of use, following the requirements of the NBR ISO/CIE 8995-1 standard [32] and the minimum specified in NR 17 [26]. Due to the specific use of the rooms that comprise a PHC facility, which hindered the analysis of the similarity of environments’ purpose of use, a minimum illuminance of 200 Lux or 300 Lux (as recommended by the illuminance scale) was considered acceptable for the reception room, whereas a minimum illuminance of 300 or 500 Lux was considered acceptable for the remaining rooms.

Based on these parameters and considering the quantitative–qualitative perception distinguishing the levels of the recommended illuminance scale, acceptable minimum and maximum illuminance levels were established to predict suitable and acceptable working conditions. Hence, the minimum acceptable illuminance in the reception rooms was established at 200 Lux, whereas the maximum acceptable level was established at 500 Lux. The minimum acceptable level for the remaining rooms was established at 300 Lux, whereas the maximum was 750 Lux.

A total of 736 indoor light levels were measured in the 23 PHC facilities, corresponding to 4 readings taken in four rooms in two shifts. Thus, illuminance was measured at four points in the rooms (Appendix A) to assess how light levels varied within the rooms at the working plane, i.e., each room was assumed to comprise four sectors. Hence, a measurement was taken at four points in the four rooms of the 23 PHC facilities in two shifts.

The lighting levels measured at each sector were used to assess whether the lighting conditions were uniform in the rooms, considering the conditions necessary for the work of PHC professionals. Hence, the rooms were assessed, and measurements were taken in the morning and afternoon. For a room to be considered suitable, it had to meet the criterion of a minimum of 70% lighting uniformity in both periods (NBR ISO/CIE 8995-1 [32]), i.e., at least three of the four measurements should be adequate in each shift [33]. If such a condition was verified in the two shifts, the room was considered adequate; if the condition was found in just one shift, the room was considered partially adequate and inadequate otherwise.

#### 2.4.4. Total Score

Finally, the total score for each PHC facility included the measurements taken in 4 rooms in 2 shifts. Thus, if the item assessed in a given room and shift was adequate, the PHC facility scored one point. Therefore, a facility’s total score ranged from zero to eight, with zero meaning that no adequate measurements were found in any of the four rooms in any shift, whereas eight means that all of the parameters were adequate. The total score was transformed from 0 to 100 with the same interpretation as when it ranged from zero to eight to facilitate understanding. Thus, a PHC facility was deemed adequate when its total score reached at least 70%.
(b)Health Professional’s Perception of the Physical Factors in the PHC Facilities

This study evaluated the perception of healthcare professionals regarding the physical risks present in primary healthcare (PHC) facilities. The concept of risk perception, which involves two aspects: the magnitude of potential loss and the probability of its occurrence, was adopted [34]. Risk perception encompasses both personal ideas and constructions and those related to the work environment because, to perceive it, you have to believe it [34]. In other words, the existence or absence of different risk factors may explain why people perceive the same risk in very different situations or why the same individual may perceive the risk differently depending on when they are asked about it [35]. This means that risk perception is subjective to the person who believes in the risk and objective to the facts of the work environment.

This study focused on the perception of physical risks, such as temperature, relative humidity, noise, and lighting, which may or may not be directly associated with the physical conditions of the work environment in PHC facilities. This study selected uncomfortable ambient temperature and humidity, annoying and irritating noise, and poor lighting as variables to correlate with the results of measurements of the physical parameters of each PHC facility to determine their percentage of adequacy. The detailed application of the concept of risk perception in research is based on a previous study [24].

The physical structure of primary healthcare (PHC) facilities plays a crucial role in determining the workers’ ability to perform their tasks effectively. The concepts of barriers and facilitators are used to evaluate the physical structure of the workplace, which are based on the International Classification of Functioning, Disability, and Health (ICF) [36]. These concepts can either promote (facilitator) or hinder (barrier) a worker’s ability to perform their duties in the PHC environment. It is important to note that the ICF theory is complex, and only a part of the concepts was used in this study. During the interview, the interviewee was asked to assess the physical structure of the PHC facility and rate it as a barrier to or a facilitator of their work. The possible responses ranged from “None”, “Mild”, “Moderate”, to “Severe”, which were adapted from the ICF reference [36].

### 2.5. Statistical Methods

The quantitative variables were described by mean and standard deviation (symmetric distribution) or median and interquartile range (asymmetric distribution), depending on the variable’s distribution, which was verified via the Shapiro–Wilk normality test. Absolute and relative frequencies described the categorical variables.

Student’s *t*-test, which compares the means of two unpaired samples, was adopted to compare the quantitative variables with symmetrical distribution; the Mann–Whitney test was used in the case of asymmetry. The Kruskal–Wallis test was used for the quantitative variables with asymmetric distribution assessed in more than two groups. Pearson’s Chi-square test was also used to analyze the adjusted residuals and assess the association between polytomous categorical variables, and the analysis of adjusted residuals was used to locate differences when the Chi-square test showed statistical significance. Finally, Fisher’s exact test assessed the association between dichotomous categorical variables; Fisher’s exact test replaced the Chi-square test when at least 25% of cells had an expected frequency of less than 5 units.

The ecological study included measurements of physical and environmental parameters (temperature, humidity, noise, and lighting) in four rooms across two shifts of 23 PHC facilities. To evaluate some associations, sociodemographic and work data and the perception of physical risk of health professionals working in these same PHC facilities were also condensed. Mean descriptive statistics for numerical variables and percentages for categorical variables were calculated for each PHC facility. In this way, it was possible to assess whether the PHC facilities with the highest rate of adequacy regarding the four criteria (Total Score) were those with the lowest rate of health professionals who perceived physical risk.

The regression or angular coefficient (b), which measures the effect on the outcome of each increase in one unit of the factor, along with the 95% confidence interval, was calculated. Furthermore, the standardized beta coefficient (β) was also presented to compare the strength of the association between the variables present in the multivariate model, as it does not have a unit of measurement, and the higher it is, the stronger the association. Finally, the coefficient of determination (R^2^) was calculated to determine the percentage of explanation of the multivariate model about a specific outcome. The criterion for entering the variable into the model was that it presented a *p*-value < 0.20 in the bivariate analysis. The significance level was set at 5% (*p* ≤ 0.05), and the analyses were performed in PRSS version 21.0 (IBM Corp., Armonk, NY, USA).

### 2.6. Ethical Considerations of the Study

All of the participants received clarifications about the study’s objectives and signed two copies of the consent form based on the Declaration of Helsinki (revised in 2013) (WMA—The World Medical). The study was initiated after approval was obtained from the Institutional Review Board (Conep) (CAAE: 70043717.0.0000.5324—Version 2—No. 2,217,759, August 2017; Version 3—No. 4,146,947, July 2020).

## 3. Results

### 3.1. Physical Factors of the Working Environments in PHC Facilities

The sample comprised 23 PHC facilities. The study’s database was organized in a matrix, with 23 lines, 1 for each facility and 598 columns, 1 for each variable. Hence, there was one line for each physical factor (parameter): air humidity, noise, illuminance, and information about the room’s conditions of use at the time of the measurement. Thus, 13,754 pieces of data were digitized and analyzed.

#### Temperature and Relative Humidity

Figure 1 shows the 184 pairs of readings concerning the temperature and relative humidity from all the rooms selected (RR, VR, CR, and PR) in each of the 23 PHC facilities, represented by the points on the diagram. The temperature readings are organized on the horizontal axis, and the relative humidity readings are on the vertical axis. Only 19 (10.3%) points, i.e., pairs of temperature and relative humidity readings, were within the recommended normative parameters [26,29].

The average temperature of the rooms was higher in the afternoon, with differences ranging between a minimum of 1.0 °C and a maximum of 1.7 °C, with an average of 1.5 °C. Note that the researchers collected data in late fall when indoor and outdoor average temperatures were around the maximum recommended temperature to ensure thermal comfort or slightly below the minimum (18 °C).

Table 1 presents the associations between the temperature and humidity (comfort) combination per room assessed during each work shift (morning and afternoon). The PHC facility in which the consultation room did not present adequate comfort levels yielded significantly higher relative humidity outdoors in the afternoon (Outdoor relative humidity, *p* = 0.013). As for the procedure room not showing adequate comfort levels, its door remained open most frequently in the afternoon (Front door open, *p* = 0.034). The adequacy criterion for thermal comfort was considered between 18 °C and 25 °C for rooms, and relative air humidity in the range between 40% and 60% was considered adequate or inadequate for values lower or higher than these.

### 3.2. Noise Levels

In this study, 368 noise levels were measured: two readings in four reference environments (RR, VR, CR, and PR) across two shifts (morning and afternoon) in the 23 PHC facilities. Figure 2 presents the average indoor noise levels, organized on a graph, where the PHC facilities appear on the horizontal axis, and the noise levels appear on the vertical axis. The four reference rooms are represented by small geometric symbols, differentiated by the rooms’ purpose of use (RR, VR, CR, and PR). The 23 vertical lines (from line 1 to line 23) correspond to the PHC facilities, each with eight records corresponding to the average noise levels taken in four rooms across two shifts.

The 184 indoor average levels are distributed between a minimum of 36.6 dB(A) and a maximum of 75.5 dB(A), with an average of 55.2 dB(A) and a standard deviation of 6.7 dB(A). Regarding the shifts, the morning shift showed an average of 55.6 dB(A) and a standard deviation of 6.6 dB(A), while an average of 54.8 dB(A) and a standard deviation of 6.7 dB(A) were found in the afternoon.

Figure 2 shows a comparison considering the maximum noise levels recommended by standard NBR 10152 (ABNT 2020), i.e., 43 (93.5%) of 46 average noise levels recorded in the reception rooms were above 50 dB(A), which is the maximum recommended. Additionally, 42 (91.3%) of 46 average noise levels recorded in the vaccination rooms were above 45 dB(A). Additionally, 44 (95.7%) of 46 average noise levels recorded in the consultation rooms were above 40 dB(A), and 45 (97.8%) of 46 levels measured in the procedure rooms were above 45 dB(A).

Table 2 presents the associations between the physical structure variables and other physical factors with noise levels according to room and shift. Note that the door of the vaccination room in the facility, which presented inadequate noise levels in the morning, remained open most of the time (Front door open, *p* = 0.021). Additionally, outdoor noise levels were significantly higher in the facility, whose consultation room presented inadequate noise levels in the morning (Movement of people, *p* = 0.016).

### 3.3. Illuminance Level

In this study, 736 lighting level values were collected in the internal environments of the services PHC monitored, corresponding to four readings in four environments, two shifts, and each of the 23 PHC facilities. Out of the 92 rooms monitored in the study, 54 (58.7%) did not have adequate lighting conditions for work. The vaccination rooms had the best adequacy, with 7 out of 23 rooms (30.4%) showing complete adequacy and another 7 (30.4%) showing partial adequacy. On the other hand, out of the 23 procedure rooms, only 5 showed partial adequacy. More details are shown in Table 3.

Table 4 presents the associations between the rooms’ physical structure and illuminance levels per room and shift. The facilities with inadequately lightened reception rooms less frequently opened internal curtains, indicating that natural light was less frequent in these working environments (Internal curtains open, *p* = 0.047).

### 3.4. Compiled Results Concerning Indoor Environments of 23 PHC Facilities

Table 5 presents the results concerning temperature and relative humidity, noise, and illuminance levels. In general, noise was the least adequate, and lighting was the most adequate parameter regardless of the room.

Figure 3 shows the percentages of the physical parameters (individual and total) deemed inadequate, partially adequate, or adequate. Noise was the parameter with the lowest percentage of adequacy (median = 0%; 25–75 percentiles: 0–16.7%), followed by the combination of temperature/humidity (median = 0%; 25–75 percentiles: 0–33.3%). The parameter with the highest percentage of adequacy was lighting (median = 25%; 25th–75th percentiles: 12.5–37.5%). The average percentage of adequacy for the three parameters in the four rooms of 23 PHC facilities was 5.6% (25–75 percentiles: 11.1–27.8%).

Considering that a percentage equal to or greater than 70% is adequate, one PHC facility presented an adequate temperature/humidity combination (4.3%), another presented adequate noise levels (4.3%), and no facility presented adequate illuminance. The general percentage shows that none of the facilities was adequate when considering the four parameters (Figure 4).

### 3.5. Health Professional’s Perception of the Physical Factors in the PHC Facilities

As previously explained, this analysis is complemented by the workers’ perceptions regarding the facilities’ physical factors, i.e., whether such factors facilitate or hinder the work performed at the PHC facilities. As shown in the method section, the results analyzed here originated from the structured database of a larger research project. This set comprises 210 PHC health professionals working in 23 facilities. The participants were 42.4 (±9.9) on average; most (84.7%) were women, married/consensual union (58.1%), and self-declared as being Caucasian (77.9%). Community health agents (47.6%), nursing technicians (20.5%), and nurses (11.9%) comprised the largest group of professionals in the sample, while most (84.3%) reported not having a second job (Appendix A).

Table 6 presents the associations between the environmental conditions of PHC facilities and the perceptions of health professionals working in them about their physical structure. There was no statistically significant association between the variables (*p* > 0.10). Additional material on condensed information from health professionals in the PHC facilities analyzed, in %, incorporated into the database of environmental conditions of the 23 PHC facilities can be found in Appendix A.

It was found that there is a significant association between the percentage of professionals who consider the physical risk moderate or severe in PHC facilities and the general percentage of the adequacy of the PHC facilities environment. This was indicated by a correlation coefficient of −0.450 and a *p*-value of 0.031, as shown in Figure 5. In other words, the lower the general percentage of the adequacy of the PHC facility’s environmental conditions, the greater the perception of moderate or severe physical risk by professionals working in the PHC facilities.

Table 7 displays the correlation between the sociodemographic and labor information of healthcare professionals employed at PHC units and the environmental conditions of the same PHC facilities. The study results indicate a statistically significant connection between the proportion of nurses working in a PHC unit and the overall adequacy of the environment (r_s_ = 0.634; *p* = 0.001). In simpler terms, the greater the percentage of nurses, the higher the percentage of adequacy of the environment in general. This finding is also depicted in Figure 6.

To account for other factors that may have influenced the results, variables that showed a *p*-value of less than 0.20 in the bivariate analysis were included in a multivariate linear regression model (Table 8). No variables were entered into the noise adequacy percentage model, and the variables entered into the lighting adequacy percentage model did not remain significant after adjustment.

After controlling for these variables, the percentage of physicians working in PHC facilities was still found to be significantly associated with the percentage of adequacy to temperature and humidity (*p* = 0.044). Together with the other variables present in the model, it explains 26.5% of the variability inherent to the outcome. Therefore, for every 1% increase in the number of physicians in PHC facilities, the adequacy of the temperature and humidity in the working environment of the PHC facilities decreases by an average of 0.95%.

Regarding general adequacy, the percentage of nurses and oral health technicians/aides remained significantly associated with this outcome after adjustment (*p* = 0.004 and *p* = 0.031, respectively). For each PHC facility with 1% more nurses, there is an average increase of 0.69% in the percentage of general adequacy (for physical parameters). For an additional 1% of oral health technicians/aides, there is an average increase of 0.74% in terms of overall adequacy.

The explanation percentage for the variables included in the general adequacy model was 60.3%, the highest of all models performed. Using the standardized regression coefficient (beta), it was observed that the variable most strongly associated with the general outcome was the percentage of nurses in the PHC facility.

## 4. Discussion

This study proposed highlighting and analyzing the physical conditions in PHC facilities by evaluating factors such as temperature, relative air humidity, noise, and illuminance. It is important to note that this study adhered to technical standards, regulations, and official guidelines to achieve its objectives. Therefore, the results of the study contribute to the understanding that examining physical parameters in the working environment of PHC services is crucial for ensuring safe, healthy, and decent working conditions [37,38].

Furthermore, it evaluated the association between health professionals’ perception regarding exposure to physical risks in the work environment of PHC facilities (subjective measures) and the adequacy of the physical parameters measured (temperature, relative humidity, noise, and illuminance levels) in the same facilities.

Due to the very nature of the activities performed in a PHC facility, the working environment and the characteristics regarding its organization, interpersonal relationships, risks arising from handling or exposure to physical factors in the working environment, and others have the potential to compromise the workers’ comfort, health, safety and performance in the short, medium, and long terms [39,40]. Hence, unsuitable working environments may also affect the quality of the care provided to the population seeking PHC services.

In this context, the combined temperature and relative humidity levels in the rooms were mainly outside the recommended normative parameters [26,29]. Nonetheless, although the criteria adopted in this study show that the temperature and relative humidity levels were not adequate, an analysis of whether the individuals were “psychologically and physiologically satisfied with the environment’s thermal conditions” [27] (p. 3) was not in the scope of this study, as it involves other environmental and personal factors.

During mild weather, the temperatures are close to the recommended thermal comfort parameters so that the doors, windows, and structure of the PHC buildings naturally maintain indoor temperatures above the outdoor temperatures, even with fluctuating outdoor temperatures resulting from the night period (not monitored here). Additionally, the doors and most windows remained open in the morning and afternoon. In this context, 100% of this study’s notes regarding the facilities’ physical structure show that outside structures comprised masonry walls, and 76% comprised concrete slabs. The construction system commonly adopted in the region involves roofs made of fiber cement tiles, which show reasonable thermal comfort performance only during mild temperatures close to the recommended parameters to ensure thermal comfort. Naturally, given the different forms of heat transfer, structures store heat, absorbing and transferring heat to the environment, reducing the impact of variations in the range of outdoor temperatures on indoor temperatures and keeping indoor environments under more stable temperature conditions [41].

The observations indicate that when the combination of temperature and relative air humidity falls outside the recommended range, it can result in inadequate thermal comfort levels for consultation and procedure rooms. This can lead to discomfort, reduced productivity, and potential health issues among the occupants, emphasizing the need for corrective measures.

However, when outside temperatures move away from mild conditions and remain lower or higher, the external walls of buildings lose thermal efficiency, and indoor temperatures begin to reflect outside temperatures: hot days reflect hot environments, whereas cold days reflect cold environments, especially at night. This situation could be minimized if materials with better thermo-hygrometric performance [42] were used; hence, such materials must be promptly specified during bidding processes for the construction or recovery of the buildings composing the physical infrastructure of PHC facilities. The strategies adopted in the facilities included the use of split-type air conditioners, greenhouses, or fans to heat or cool the environments by circulating the air. For instance, buildings can be equipped to use natural energy, an essential aspect of environmental health.

The ecological study of professionals’ risk perception regarding exposure to physical factors in their PHC work environment highlighted the fact that physicians tend to perceive more physical risk from temperature and humidity. This fact was observed in the correlation association analysis between the increase in the number of physicians in PHC facilities and the lower adequacy of the temperature and humidity parameters measured in the PHC facilities; however, these results should be interpreted with caution. This is likely related to the conditions of the monitored environments in relation to the consultation room, where medical professionals spend several working hours.

A literature review addressing studies investigating indoor environments indicates that low temperatures, i.e., below the thermal comfort parameters, increase the risk of cardiovascular and respiratory diseases. In contrast, high temperatures increase the risk of acute symptoms, e.g., dry eyes and respiratory symptoms [1]. This finding suggests a need to investigate people’s psychological and physiological satisfaction with the working environment’s thermal conditions. Hence, this is an important conceptual and operational element for future studies addressing these facilities.

Therefore, these results suggest the need to reevaluate or re-plan the procedures in the daily work of PHC facilities to improve the existing infrastructure. Additionally, this study’s limitations concerning temperature and relative humidity are related to normative requirements established for air-conditioned environments. These parameters were extended to the analysis of PHC environments to verify whether they were being adopted during data collection. Furthermore, considering that this study was carried out during the period of restrictions due to the COVID-19 pandemic, the researchers assumed that the conditions of use of the environments discussed here incorporate national requirements [43], regional, or international requirements [44] established for each type of environment and other contingent guidelines on the use of spaces and the crowing of people, such as keeping the PHC rooms ventilated by opening the doors and windows whenever possible.

The study found that noise was the least adequate factor in terms of physical comfort in PHC working environments, regardless of the room (reception room (SR), vaccination room (SV), consultation room (SC), or technical procedures room (SP)). Notably, in the vaccination room, PHC facilities with open entrance doors had higher noise levels in the morning. In the consultation room, PHC facilities with inadequate noise had significantly higher external noise levels in the morning. Additionally, reception rooms had the highest indoor average noise levels in the morning and afternoon, and their outdoor noise levels were also among the highest compared to the other three environments (SV, SC, and SP). It is indispensable that this issue be addressed promptly to ensure the well-being of both staff and patients.

The term noise “encompasses all unwanted or disturbing sound with unpleasant acoustic characteristics that may result in hearing impairment or be harmful to health or otherwise dangerous” [18] (p. 332). Therefore, the higher noise levels can be explained by the conditions of use of reception rooms; 93.5% of these kept the front doors open, and 50.0% kept the windows open. Under these conditions of use, there is a crowd and movement of people seeking care at the PHC facility who remain in its immediate surroundings. Because data were collected during the COVID-19 pandemic, when there was the risk of SARC-CoV-2 transmission, the services were instructed to keep outside doors and windows open whenever possible to ventilate the environment [43,44]. Furthermore, people waiting to schedule appointments or other services were not allowed to enter the building; only those receiving care were allowed.

Considering the guiding technical context, the level of background noise pressure that ensures comfort, according to standard NR 17, must comply with the reference parameters for indoor environments according to their purpose of use established in official technical standards. The acceptable background noise levels for acoustic comfort for the remaining cases are up to 65 dB(A) [26]. The NBR 10152 standard—Acoustics—establishes reference values for noise sound pressure levels in indoor environments [30].

The ambience of PHC facilities is where health services and the population meet, a locus of social relationships that transcends objective professional actions, ensuring access and safety and welcoming people, their families, their habits, and values [45]. Therefore, it is a space convergent with conversations, children’s crying, cell phone use, music, and the barking of dogs that accompany their owners and wait for them in front of the facility. There are also noises coming from the traffic, equipment, and, finally, noises that characterize the surroundings of PHC facilities and contribute to the acoustic levels in their environments. Such a condition is aggravated by the constant need to ventilate the environment by opening doors and windows whenever possible.

Therefore, given the naturalization of noise in PHC facilities, acoustic comfort conditions may not be sufficiently valued. Immediately minimizing this problem requires organizing and managing work to implement procedures to reduce noise, such as optimizing the flow of people and waiting time, educating people by using different communication strategies [6], asking people to remain silent, and performing preventive maintenance on the unit’s equipment (air conditioners, refrigerators, fans, motors, and other sources of noise) [46].

However, indoor noise reference levels must be established over time, considering the specific actions typically performed in PHC facilities, qualifying new or rebuilding projects. Computer simulation, with or without Artificial Intelligence, currently estimates how noise propagates in different architectural projects, indicating the best innovative materials, construction techniques, construction systems, and acoustic barriers to ensure efficient building projects within PHC facilities [29]. It is noteworthy that good lighting in working environments involves aspects, procedures, and assessments of requirements outside the scope of this study [32].

Illuminance was the parameter presenting the highest level of adequacy, regardless of the room. However, according to the criterion adopted here, a minimum of 70% illuminance was required to indicate adequate lighting comfort [32], and none of the facilities met such criterion. Naturally, poor illuminance results from deficient artificial lighting, which, when adequate and effectively activated, ensures minimum lighting at acceptable levels; such an affirmation is supported by the conditions in which the rooms were found. At the other extreme, levels above the maximum acceptable parameters were found in the PHC facilities. Such a condition is determined by direct natural light entering the room through windows with open curtains or without curtains. These environments may lead to problems such as undue shading, reflective or direct glare, or other inadequacies, hindering the work routine, with consequent risks to the health workers’ visual and psychophysiological health [47,48,49].

This study’s findings highlighted trends in visual comfort in the working environment by comparing the rooms and the PHC facilities. It is worth noting that adequately maintaining and restoring PHC facilities, in the short or medium term, can minimize the deficiencies in the physical environment regarding lighting. In this sense, improving illuminance and ensuring it meets the minimum recommended levels in working environments can be achieved by using artificial lighting (lamps) and adjustable curtains on windows to block (otherwise desirable) natural light when needed [49,50].

It is important to note that evidence shows a relationship between the general adequacy percentage of monitored parameters and specific attributes included in a multivariate linear regression model. Specifically, there is a connection between the general adequacy percentage of physical parameters (objective measures) and the perception of physical risk (subjective measure) reported by healthcare professionals working in the same monitored PHC facilities. In other words, the higher the level of perceived moderate or severe physical risk by healthcare professionals, the lower the general percentage of adequacy of the working environment conditions of the evaluated parameters.

Furthermore, this study found an association between the number of nurses and oral health assistants and the general level of adequacy of the physical parameters evaluated in the PHC facilities. This means that the greater the number of nurses, the higher the general percentage of the adequacy of the monitored parameters concerning environmental work comfort. The associations represent areas that deserve further study in future research. 

However, it indicates that the increase in nurses and the presence of oral health assistants are related to PHC facilities with more than one multidisciplinary PHC team. Furthermore, they have a physical structure with larger dimensions and are structured for expanded teams, such as the presence of a dentist, for example. However, remembering the specific focus of the medical professional, there was a different result; the increase in this professional tended to reduce the adequacy percentage concerning these specific parameters (temperature/humidity). As previously mentioned, it may be linked to the particular stay of these medical professionals in these rooms in the PHC facility.

It is essential to evaluate the buildings that comprise a network of healthcare outpatient clinics since each primary healthcare (PHC) facility is part of a larger structural body. Doing so can ensure the sustainability of human health [51]. This evaluation will help workers and managers understand the relevance of other investigations that may arise from this local study to their daily work lives. Additionally, it is crucial to develop theoretical frameworks that can help us understand the impacts of physical environments on human health and well-being, as hazardous physical factors in workplaces can negatively affect both humans and the surrounding nature [52].

Perception is a subjective experience that involves an individual’s understanding of occupational risks and the objective characteristics of their working conditions. Knowledge about how people perceive risks is essential for developing strategies to improve the physical working conditions that promote the well-being of individuals and the collective workforce in the primary healthcare environment.

### Limitations and Lines of Research

It is noteworthy that among the limitations of this study is that the cross-sectional design does not allow for causal relationships to be established. Therefore, future studies should include longitudinal designs concerning objective metrics, physical factors of the work environment, and the perception of health professionals about the risk of physical factors that negatively or positively influence physical comfort and, consequently, the development of work and development application of collective and technically sustainable strategies to change the physical conditions of the work environment in PHC facilities. Furthermore, although an adequate sample size was adopted, intentional sampling was used for health professionals in the PHC facilities selected to study physical factors because it was carried out in only one municipality. Therefore, caution is needed in interpreting and generalizing the results.

Another limitation is that only some studies address this work environment at PHC facilities, which prevents comparisons between the physical factors of the work environment and workers’ perceptions in a historical series of changes in working conditions. In this sense, this study can guide future studies and support an in-depth analysis of the movement of workers and managers toward improvements in the physical conditions of the work environment at PHC facilities.

## 5. Conclusions

The current research demonstrated a low prevalence of adequacy of the evaluated physical factors (temperature, relative humidity, noise, and illuminance levels) in the four rooms (reception room (RR), vaccination room (VR), consultation room (CR), and technical procedures room (RP)) and both shifts (morning and afternoon) in the 23 PHC facilities, that require for improvement. Noise was the least appropriate factor, highlighting an area that requires immediate attention. Even with low values, illuminance emerged as the most suitable parameter regardless of the environment.

Revisiting the central question of this study, an association was confirmed between the risk perception of health professionals working in the PHC service and the physical environmental conditions of the same PHC facilities, notably when considering the physical parameters evaluated. Essentially, a lower overall percentage of the adequacy of physical parameters assessed in PHC facilities corresponds to a higher perception of moderate or severe physical risk among health professionals who work there. Furthermore, specific trends were identified, such as when more doctors are in PHC facilities, the temperature and humidity adequacy percentage tends to decrease. On the other hand, when there are more nurses and oral health assistants in a PHC unit, there is a higher percentage of the adequacy of the monitored physical parameters included in possible physical comfort in the work environment.

The inclusion of health professionals’ perception of physical risk about objective measurements of the physical parameters of the monitored PHC facilities confirmed the expectation that studies encompassing these two data sources validate the necessary claims for improvements in the physical comfort of the working environment in PHC facilities. In other words, workers’ perceptions express their workplace’s physical conditions to direct collective strategies toward the necessary changes in PHC facilities.

Thus, the evidence presented in this study contributes to deepening understanding and decision-making regarding the physical conditions addressed here by focusing on objective assessments and the workers’ subjective perceptions. From a broader perspective, one must consider that physical factors (temperature and relative humidity, noise, and illuminance levels) are examples of (positive/negative) conditions determining healthy, safe, and decent work. In this sense, these results guide and encourage the development of other studies adopting longitudinal methodologies and occupational interventions to monitor working environments to ensure the comfort, safety, and well-being of PHC workers.

## Figures and Tables

**Figure 1 ijerph-21-00847-f001:**
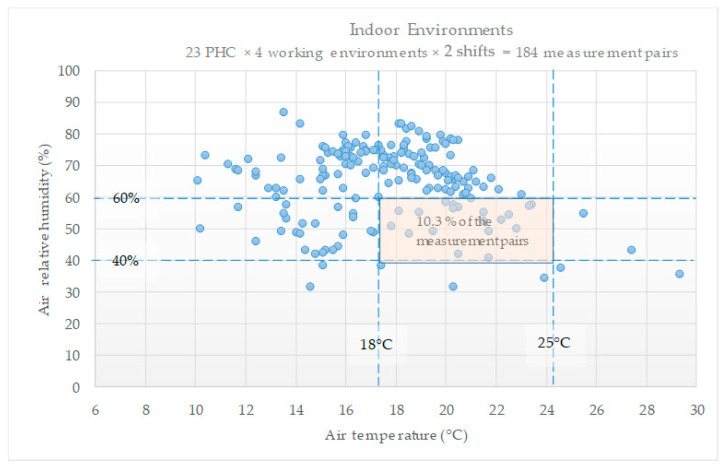
Temperature and relative humidity in PHC facilities.

**Figure 2 ijerph-21-00847-f002:**
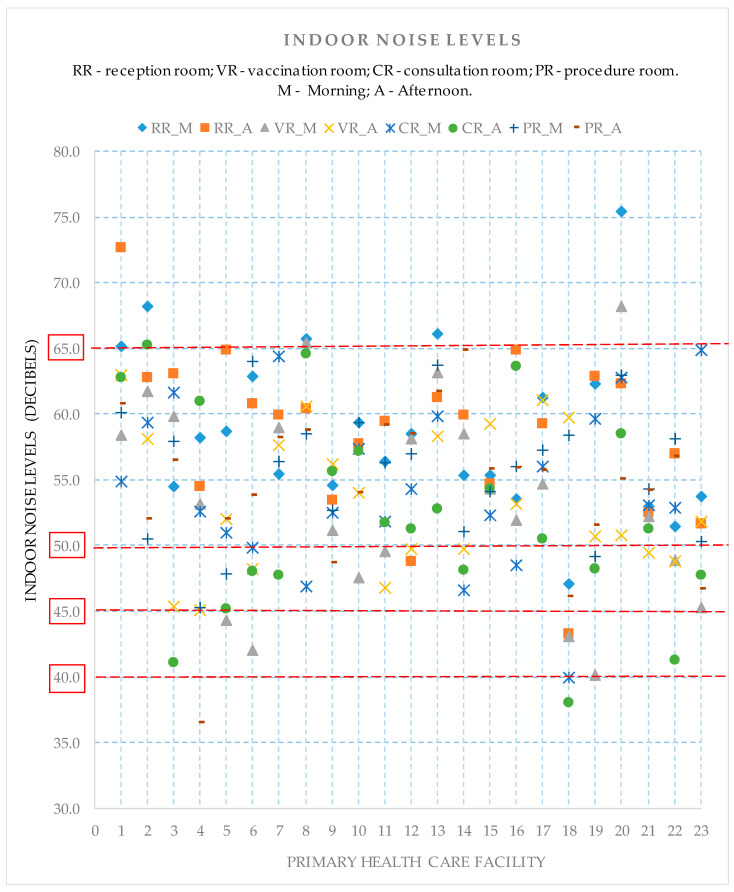
Average noise levels were measured in each PHC facility (from line 1 to line 23), with eight records in each of four rooms and across two shifts.

**Figure 3 ijerph-21-00847-f003:**
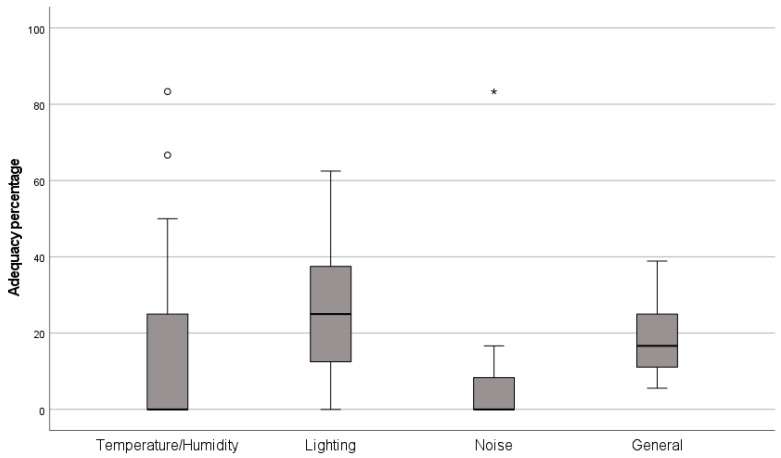
Percentage of adequacy levels (individual and total) obtained in 23 PHC facilities. The vertical line in the box plot represents the median, and the lower and upper limits represent the first and third quartiles. Circles (°) and asterisks (*) represent extreme values in the sample. Circles exceeded 1.5 times the interquartile range and asterisks exceeded 3 times this value.

**Figure 4 ijerph-21-00847-f004:**
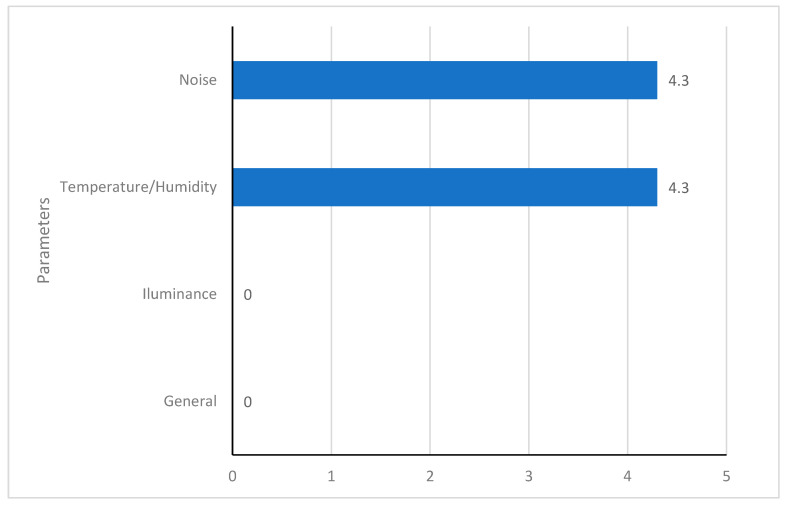
Percentage of PHC facilities with adequacy levels equal to or above 70% in three parameters.

**Figure 5 ijerph-21-00847-f005:**
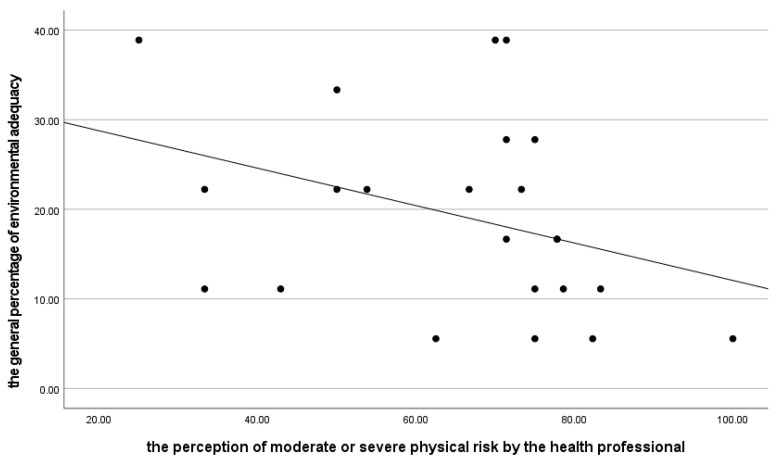
Association between the general percentage of environmental adequacy and the perception of moderate or severe physical risk by the health professional.

**Figure 6 ijerph-21-00847-f006:**
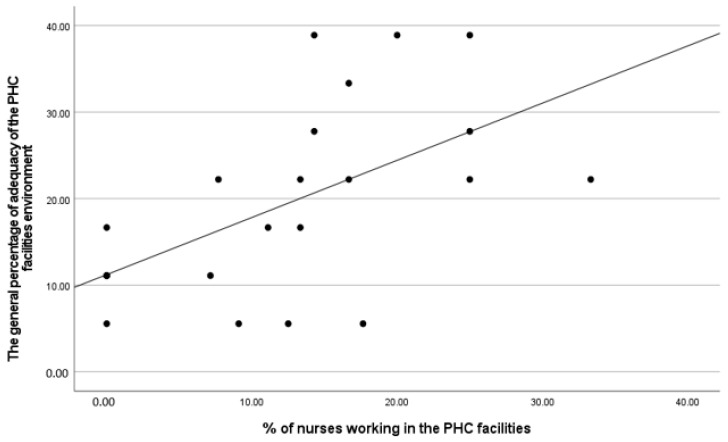
Association between the general percentage of environmental adequacy and the percentage of nurses working in the PHC facilities.

**Table 1 ijerph-21-00847-t001:** Association with the temperature and humidity combination, the physical characteristics of the PHC facilities and their surroundings, in the morning and afternoon shifts.

Room	Variables	Morning ^‡^				Afternoon ^‡^			
		n (%)	Not Adequate	Adequate	*p*	n (%)	Not Adequate	Adequate	*p*
			n (%)	n (%)			n (%)	n (%)	
Reception room	Total	23 (100)	22 (95.7)	1 (4.3)		23 (100)	20 (87.0)	3 (13.0)	
	Lightweight partition wall	4 (17.4)	3 (13.6)	1 (100)	0.174 ^a^	4 (17.4)	3 (15.0)	1 (33.3)	0.453 ^a^
	Concrete slab	18 (78.3)	17 (77.3)	1 (100)	1.000 ^a^	18 (78.3)	15 (75.0)	3 (100)	1.000 ^a^
	Lightweight ceiling panels	5 (21.7)	5 (22.7)	0 (0.0)	1.000 ^a^	5 (21.7)	5 (25.0)	0 (0.0)	1.000 ^a^
	Front door open	22 (95.7)	21 (95.5)	1 (100)	1.000 ^a^	22 (95.7) ^a^	19 (95.0)	3 (100)	1.000 ^a^
	Outside window open	13 (56.5)	13 (59.1)	0 (0.0)	0.435 ^a^	9 (39.1) ^a^	7 (35.0)	2 (66.7)	0.538 ^a^
	Ceiling light on	21 (91.3)	21 (95.5)	0 (0.0)	0.087 ^a^	20 (87.0)	17 (85.0)	3 (100)	1.000 ^a^
	Sunny weather	10 (43.5)	9 (40.9)	1 (100)	0.435 ^a^	10 (43.5)	7 (35.0)	3 (100)	0.068 ^a^
	Cloudy weather	8 (34.8)	8 (36.4)	0 (0.0)	1.000 ^a^	6 (26.1)	6 (30.0)	0 (0.0)	0.539 ^a^
	Outdoor temperature *	15.6 ± 3.1	15.5 ± 3.1	17.2 ± 0.0	0.605 ^b^	17.2 ± 3.4	16.7 ± 3.4	20.3 ± 2.5	0.093 ^b^
	Relative humidity *	73.0 ± 12.3	72.9 ± 12.6	74.0 ± 0.0	0.935 ^b^	65.0 ± 11.0	65.7 ± 11.3	60.1 ± 8.6	0.428 ^b^
	Outdoor wind speed **	1.2 (0.2–2)	1.1 (0.2–2.0)	1.2 (1.2–1.2)	0.783 ^c^	1.7 (0.2–2.3)	1.7 (0.6–2.4)	0.0 (0.0–0.0)	0.115 ^c^
Vaccination room	Total	23 (100)	21 (91.3)	2 (8.7)		23 (100)	19 (82.6)	4 (17.4)	
	Lightweight partition wall	1 (4.3)	1 (4.8)	0 (0.0)	1.000 ^a^	1 (4.3)	1 (5.3)	0 (0.0)	1.000 ^a^
	Concrete slab	18 (78.3)	17 (81.0)	1 (50.0)	0.395 ^a^	18 (78.3)	15 (78.9)	3 (75.0)	1.000 ^a^
	Lightweight ceiling panels	5 (21.7)	4 (19.0)	1 (50.0)	0.395 ^a^	5 (21.7)	4 (21.1)	1 (25.0)	1.000 ^a^
	Front door open	18 (78.3)	17 (81.0)	1 (50.0)	0.395 ^a^	19 (82.6)	17 (89.5)	2 (50.0)	0.125 ^a^
	Outside window open	8 (34.8)	7 (33.3)	1 (50.0)	1.000 ^a^	8 (34.8)	6 (31.6)	2 (50.0)	0.589 ^a^
	Ceiling light on	20 (87.0)	18 (85.7)	2 (100)	1.000 ^a^	21 (91.3)	18 (94.7)	3 (75.0)	0.324 ^a^
	Air-conditioned on	4 (17.4)	3 (14.3)	1 (50.0)	0.324 ^a^	4 (17.4)	3 (15.8)	1 (25.0)	1.000 ^a^
	Autoclave on	3 (13.0)	3 (14.3)	0 (0.0)	1.000 ^a^	4 (17.4)	4 (21.1)	0 (0.0)	1.000 ^a^
	Sunny weather	10 (43.5)	9 (42.9)	1 (50.0)	1.000 ^a^	11 (47.8)	8 (42.1)	3 (75.0)	0.317 ^a^
	Cloudy weather	7 (30.4)	7 (33.3)	0 (0.0)	1.000 ^a^	5 (21.7)	5 (26.3)	0 (0.0)	0.539 ^a^
	Outdoor temperature *	15.6 ± 3.0	15.3 ± 3.0	18.1 ± 2.7	0.227 ^b^	17.1 ± 3.0	16.8 ± 3.0	18.7 ± 3.0	0.267 ^b^
	Outdoor relative humidity *	71.8 ± 13.0	71.8 ± 13.7	71.7 ± 3.3	0.989 ^b^	64.9 ± 11.0	65.5 ± 10.7	61.9 ± 13.6	0.568 ^b^
	Outdoor wind speed **	0.9 (0.0–1.6)	0.9 (0.0–1.4)	2.1 (1.6–2.6)	0.158 ^c^	0.4 (0.0–2.1)	0.1 (0.0–2.1)	1.3 (0.5–3.2)	0.250 ^c^
Consultation room	Total	23 (100)	23 (100)	0 (0.0)		23 (100)	19 (82.6)	4 (17.4)	
	Lightweight partition wall	3 (13.0)	3 (13.0)			3 (13.0)	2 (10.5)	1 (25.0)	0.453 ^a^
	Concrete slab	15 (65.2)	15 (65.2)			15 (65.2)	12 (63.2)	3 (75.0)	1.000 ^a^
	Lightweight ceiling panels	8 (34.8)	8 (34.8)			8 (34.8)	7 (36.8)	1 (25.0)	1.000 ^a^
	Front door open	19 (82.6)	19 (82.6)			20 (87.0)	17 (89.5)	3 (75.0)	0.453 ^a^
	Outside window open	7 (30.4)	7 (30.4)			7 (30.4)	6 (31.6)	1 (25.0)	1.000 ^a^
	Ceiling light on	19 (82.6)	19 (82.6)			15 (65.2)	12 (63.2)	3 (75.0)	1.000 ^a^
	Air-conditioned on	1 (4.3)	1 (4.3)						
	Autoclave on	5 (21.7)	5 (21.7)			2 (8.7)	1 (5.3)	1 (25.0)	0.324 ^a^
	Sunny weather	10 (43.5)	10 (43.5)			11 (47.8)	8 (42.1)	3 (75.0)	0.317 ^a^
	Cloudy weather	6 (26.1)	6 (26.1)			6 (26.1)	6 (31.6)	0 (0.0)	0.539 ^a^
	Outdoor temperature *	15.4 ± 3.3	15.4 ± 3.3			17.1 ± 3.1	16.8 ± 3.2	18.4 ± 1.9	0.381 ^b^
	Outdoor relative humidity *	73.1 ± 10.5	73.1 ± 10.5			66.4 ± 10.5	68.8 ± 9.9	54.9 ± 3.2	0.013 ^b^
	Outdoor wind speed **	0.7 (0.0–1.7)	0.7 (0.0–1.7)			0.7 (0.0–1.7)	0.6 (0.0–2.0)	0.8 (0.6–0.8)	0.969 ^c^
Procedure room	Total	23 (100)	21 (91.3)	2 (8.7)		23 (100)	20 (87.0)	3 (13.0)	
	Lightweight partition wall	3 (13.0)	3 (14.3)	0 (0.0)	1.000 ^a^	3 (13.0)	3 (15.0)	0 (0.0)	1.000 ^a^
	Concrete slab	17 (73.9)	15 (71.4)	2 (100)	1.000 ^a^	17 (73.9)	14 (70.0)	3 (100)	0.539 ^a^
	Lightweight ceiling panels	6 (26.1)	6 (28.6)	0 (0.0)	1.000 ^a^	6 (26.1)	6 (30.0)	0 (0.0)	0.539 ^a^
	Front door open	21 (91.3)	19 (90.5)	2 (100)	1.000 ^a^	20 (87.0)	19 (95.0)	1 (33.3)	0.034 ^a^
	Outside window open	11 (47.8)	9 (42.9)	2 (100)	0.217 ^a^	12 (52.2)	9 (45.0)	3 (100)	0.217 ^a^
	Ceiling light on	17 (73.9)	16 (76.2)	1 (50.0)	0.462 ^a^	18 (78.3)	16 (80.0)	2 (66.7)	0.539 ^a^
	Air-conditioned on	1 (4.3)	1 (4.8)	0 (0.0)	1.000 ^a^	1 (4.3)	1 (5.0)	0 (0.0)	1.000 ^a^
	Refrigerator on	2 (8.7)	2 (9.5)	0 (0.0)	1.000 ^a^	2 (8.7)	2 (10.0)	0 (0.0)	1.000 ^a^
	Sunny weather	9 (39.1)	8 (38.1)	1 (50.0)	1.000 ^a^	9 (39.1)	7 (35.0)	2 (66.7)	0.538 ^a^
	Cloudy weather	9 (39.1)	8 (38.1)	1 (50.0)	1.000 ^a^	6 (26.1)	5 (25.0)	1 (33.3)	1.000 ^a^
	Outdoor temperature *	15.6 ± 3.1	15.6 ± 3.1	16.3 ± 3.4	0.750 ^b^	16.6 ± 4.2	16.4 ± 4.5	18.3 ± 1.2	0.483 ^b^
	Outdoor wind speed **	1.4 (0.0–1.9)	1.2 (0.0–2.0)	1.4 (1.4–1.4)	0.870 ^c^	0.8 (0.0–1.5)	0.7 (0.0–1.7)	0.8 (0.8–1.5)	0.573 ^c^

* Described by mean ± SD; ** described by median (percentiles 25–75); ^a^ Fisher’s Exact test; ^b^ student’s *t*-test; ^c^ Mann–Whitney test; **^‡^** Zone Time: UTC-3 BRT (Brasília Time); Adequate = temperature between 18 °C–25 °C and relative humidity between 40% and 60%. Inadequate = values above or below-indicated values.

**Table 2 ijerph-21-00847-t002:** Association with noise levels, the physical characteristics of the PHC facilities and their surroundings, in the morning and afternoon shifts.

Room	Variables	Morning ^‡^				Afternoon ^‡^			
		n (%)	Not Adequate	Adequate	*p*	n (%)	Not Adequate	Adequate	*p*
			n (%)	n (%)			n (%)	n (%)	
Reception room	Total	23 (100)	22 (95.7)	1 (4.3)		23 (100)	21 (91.3)	2 (8.7)	
	Lightweight partition wall	4 (17.4)	4 (18.2)	0 (0.0)	1.000 ^a^	4 (17.4)	3 (14.3)	1 (50.0)	0.324 ^a^
	Concrete slab	18 (78.3)	17 (77.3)	1 (100)	1.000 ^a^	18 (78.3)	16 (76.2)	2 (100)	1.000 ^a^
	Lightweight ceiling panels	5 (21.7)	5 (22.7)	0 (0.0)	1.000 ^a^	5 (21.7)	5 (23.8)	0 (0.0)	1.000 ^a^
	Front door open	21 (91.3)	20 (90.9)	1 (100)	1.000 ^a^	22 (95.7)	21 (100)	1 (50.0)	0.087 ^a^
	Outside window open Air-conditioned	13 (56.5)	12 (54.5)	1 (100)	1.000 ^a^	10 (43.5)	9 (42.9)	1 (50.0)	1.000 ^a^
	on					1 (4.3)	1 (4.8)	0 (0.0)	1.000 ^a^
	Outdoor noise *	56.5 ± 6.1	56.9 ± 5.9	48.7 ± 0.0	0.193 ^b^	65.0 ± 11.0	58.3 ± 5.9	57.5 ± 9.3	0.868 ^b^
	Movement of people	21 (91.3)	21 (95.5)	0 (0.0)	0.087 ^a^	20 (87.0)	19 (90.5)	1 (50.0)	0.249 ^a^
	People talking	13 (56.5)	12 (54.5)	1 (100)	1.000 ^a^	13 (56.5)	11 (52.4)	2 (100)	0.486 ^a^
	Motorcycle traffic	17 (73.9)	16 (72.7)	1 (100)	1.000 ^a^	14 (60.9)	14 (66.7)	0 (0.0)	0.142 ^a^
	Light vehicle traffic	18 (78.3)	18 (81.8)	0 (0.0)	0.217 ^a^	21 (91.3)	20 (95.2)	1 (50.0)	0.170 ^a^
	Heavy vehicle traffic	8 (34.8)	8 (36.4)	0 (0.0)	1.000 ^a^	7 (30.4)	7 (33.3)	0 (0.0)	1.000 ^a^
	School noise	8 (34.8)	8 (36.4)	0 (0.0)	1.000 ^a^	6 (26.1)	6 (28.6)	0 (0.0)	1.000 ^a^
	Commercial noise	2 (8.7)	2 (9.1)	0 (0.0)	1.000 ^a^	3 (13.0)	3 (14.3)	0 (0.0)	1.000 ^a^
	Noise from workshops					2 (8.7)	2 (9.5)	0 (0.0)	1.000 ^a^
	Rain noise	1 (4.3)	1 (4.5)	0 (0.0)	1.000 ^a^	1 (4.3)	1 (4.8)	0 (0.0)	1.000 ^a^
	Wind noise	9 (39.1)	9 (40.9)	0 (0.0)	1.000 ^a^	10 (43.5)	9 (42.9)	1 (50.0)	1.000 ^a^
Vaccination room	Total	23 (100)	19 (82.6)	4 (17.4)		23 (100)	23 (100)	0 (0.0)	
	Lightweight partition wall	1 (4.3)	1 (5.3)	0 (0.0)	1.000 ^a^	1 (4.3)	1 (4.3)		
	Concrete slab	18 (78.3)	16 (84.2)	2 (50.0)	0.194 ^a^	18 (78.3)	18 (78.3)		
	Lightweight ceiling panels	5 (21.7)	3 (15.8)	2 (50.0)	0.194 ^a^	5 (21.7)	5 (21.7)		
	Front door open	18 (78.3)	17 (89.5)	1 (25.0)	0.021 ^a^	19 (82.6)	19 (82.6)		
	Outside window open Air-conditioned	8 (34.8)	7 (36.8)	1 (25.0)	1.000 ^a^	6 (26.1)	6 (26.1)		
	on	4 (17.4)	2 (10.5)	2 (50.0)	0.125 ^a^	4 (17.4)	4 (17.4)		
	Outdoor noise *	57.8 ± 6.1	58.0 ± 6.2	56.5 ± 6.7	0.654 ^b^	55.3 ± 6.4	55.3 ± 6.4		
	Movement of people	16 (69.6)	14 (73.7)	2 (50.0)	0.557 ^a^	19 (82.6)	19 (82.6)		
	People talking	13 (56.5)	10 (52.6)	3 (75.0)	0.604 ^a^	13 (56.5)	13 (56.5)		
	Motorcycle traffic	14 (60.9)	10 (52.6)	4 (100)	0.127 ^a^	14 (60.9)	14 (60.9)		
	Light vehicle traffic	15 (65.2)	12 (63.2)	3 (75.0)	1.000 ^a^	17 (73.9)	17 (73.9)		
	Heavy vehicle traffic	9 (39.1)	8 (42.1)	1 (25.0)	1.000 ^a^	7 (30.4)	7 (30.4)		
	School noise	8 (34.8)	7 (36.8)	1 (25.0)	1.000 ^a^	6 (26.1)	6 (26.1)		
	Commercial noise	3 (13.0)	3 (15.8)	0 (0.0)	1.000 ^a^	3 (13.0)	3 (13.0)		
	Noise from workshops	1 (4.3)	1 (5.3)	0 (0.0)	1.000 ^a^	2 (8.7)	2 (8.7)		
	Rain noise	1 (4.3)	1 (5.3)	0 (0.0)	1.000 ^a^				
	Wind noise	7 (30.4)	7 (36.8)	0 (0.0)	0.273 ^a^	7 (30.4)	7 (30.4)		
Consultation room	Total	23 (100)	22 (95.7)	1 (4.3)		23 (100)	22 (95.7)	1 (4.3)	
	Lightweight partition wall	3 (13.0)	3 (13.6)	0 (0.0)	1.000 ^a^	3 (13.0)	3 (13.6)	0 (0.0)	1.000 ^a^
	Concrete slab	15 (65.2)	15 (68.2)	0 (0.0)	0.348	15 (65.2)	15 (68.2)	0 (0.0)	0.348 ^a^
	Lightweight ceiling panels	8 (34.8)	7 (31.8)	1 (100)	0.348 ^a^	8 (34.8)	7 (31.8)	1 (100)	0.348 ^a^
	Front door open	18 (78.3)	17 (77.3)	1 (100)	1.000 ^a^	20 (87.0)	19 (86.4)	1 (100)	1.000 ^a^
	Outside window open Air-conditioned	6 (26.1)	6 (27.3)	0 (0.0)	1.000 ^a^	7 (30.4)	7 (31.8)	0 (0.0)	1.000 ^a^
	on	1 (4.3)	1 (4.5)	0 (0.0)	1.000 ^a^				
	Outdoor noise *	1 (4.3)	1 (4.5)	0 (0.0)	1.000 ^a^				
	Movement of people	53.9 ± 5.7	54.5 ± 5.0	41.0 ± 0.00	0.016 ^b^	55.4 ± 6.4			
	People talking	16 (69.6)	16 (72.7)	0 (0.0)	0.304 ^a^	17 (73.9)	16 (72.7)	1 (100)	1.000 ^a^
	Motorcycle traffic	12 (52.2)	11 (50.0)	1 (100)	1.000 ^a^	10 (43.5)	9 (40.9)	1 (100)	0.435 ^a^
	Light vehicle traffic	11 (47.8)	11 (50.0)	0 (0.0)	1.000 ^a^	12 (52.2)	12 (54.5)	0 (0.0)	0.478 ^a^
	Heavy vehicle traffic	13 (56.5)	13 (59.1)	0 (0.0)	0.435 ^a^	15 (65.2)	15 (68.2)	0 (0.0)	0.348 ^a^
	School noise	8 (34.8)	8 (36.4)	0 (0.0)	1.000 ^a^	7 (30.4)	7 (31.8)	0 (0.0)	1.000 ^a^
	Commercial noise	6 (26.1)	6 (27.3)	0 (0.0)	1.000 ^a^	5 (21.7)	5 (22.7)	0 (0.0)	1.000 ^a^
	Noise from workshops	3 (13.0)	3 (13.6)	0 (0.0)	1.000 ^a^	3 (13.0)	3 (13.6)	0 (0.0)	1.000 ^a^
	Rain noise	8 (34.8)	8 (36.4)	0 (0.0)	1.000 ^a^	11 (47.8)	11 (50.0)	0 (0.0)	1.000 ^a^
Procedures room	Total	23 (100)	23 (100)	0 (0.0)		23 (100)	22 (95.7)	1 (4.3)	
	Lightweight partition wall	3 (13.0)	3 (13.0)			3 (13.0)	2 (9.1)	1 (100)	0.130 ^a^
	Concrete slab	17 (73.9)	17 (73.9)			17 (73.9)	16 (72.7)	1 (100)	1.000 ^a^
	Lightweight ceiling panels	6 (26.1)	6 (26.1)			6 (26.1)	6 (27.3)	0 (0.0)	1.000 ^a^
	Front door open	21 (91.3)	21 (91.3)			19 (82.6)	18 (81.8)	1 (100)	1.000 ^a^
	Outside window open Air-conditioned	12 (52.2)	12 (52.2)			10 (43.5)	10 (45.5)	0 (0.0)	1.000 ^a^
	on	1 (4.3)	1 (4.3)			1 (4.3)	1 (4.5)	0 (0.0)	1.000 ^a^
	Outdoor noise *					1 (4.3)	1 (4.5)	0 (0.0)	1.000 ^a^
	Movement of people	2 (8.7)	2 (8.7)			2 (8.7)	2 (9.1)	0 (0.0)	1.000 ^a^
	People talking	54.4 ± 4.8	54.4 ± 4.8			55.9 ± 8.7	55.3 ± 6.5	58.8 ± 0.0	0.601 ^b^
	Motorcycle traffic	19 (82.6)	19 (82.6)			20 (87.0)	19 (86.4)	1 (100)	1.000 ^a^
	Light vehicle traffic	12 (52.2)	12 (52.2)			10 (43.5)	10 (45.5)	0 (0.0)	1.000 ^a^
	Heavy vehicle traffic	14 (60.9)	14 (60.9)			11 (47.8)	11 (50.0)	0 (0.0)	1.000 ^a^
	School noise	13 (56.5)	13 (56.5)			16 (69.6)	15 (68.2)	1 (100)	1.000 ^a^
	Commercial noise	9 (39.1)	9 (39.1)			8 (34.8)	8 (36.4)	0 (0.0)	1.000 ^a^
	Noise from workshops	6 (26.1)	6 (26.1)			5 (21.7)	4 (18.2)	1 (100)	0.217 ^a^
	Rain noise	4 (17.4)	4 (17.4)			4 (17.4)	3 (13.6)	1 (100)	0.174 ^a^
	Wind noise	13 (56.5)	13 (56.5)			12 (52.2)	12 (54.5)	0 (0.0)	0.478 ^a^

* Described by mean ± DP; ^a^ Fisher’s Exact test; ^b^ student’s *t*-test; ^‡^ Zone Time: UTC-3 BRT (Brasília Time); Adequate = reception room—50 dB(A); vaccination and procedure rooms—45 dB(A); and consultation room—40 dB(A); Not adequate = values above those indicated.

**Table 3 ijerph-21-00847-t003:** Number and percentage of rooms according to work shifts in which illuminance was assessed as being adequate at work.

Rooms (Minimum and Maximum Acceptable Illuminance Levels)	Rooms in Condition Illuminance Adequate for Work						Total (Rooms)	
	2 Shifts ^‡^		1 Shift ^‡^		No Shift ^‡^			
	n	%	n	%	n	%	n	%
Reception rooms—sr (≥200 Lux and <500 Lux)	4	17.4	8	34.8	11	47.8	23	100
Vaccination rooms—sv (≥300 Lux; <750 Lux)	7	30.4	7	30.4	9	39.1	23	100
Consultation rooms—sc (≥300 Lux; <750 Lux)	1	4.3	6	26.0	16	69.6	23	100
Procedure rooms—sp (≥300 Lux; <750 Lux)	0	0	5	21.7	18	78.3	23	100
Total	12	13.0	26	28.3	54	58.7	92	100

Recommended minimum illuminance levels: sr = 200 Lux and sv = sc = sp = 300 Lux; maximum acceptable illuminance levels: sr = 500 Lux and sv = sc = sp = 750 Lux; n or %–number or percentage of environments in illuminance adequate for work. ^‡^ Zone Time: UTC-3 BRT (Brasília Time).

**Table 4 ijerph-21-00847-t004:** Association with illuminance levels, the physical characteristics of the PHC facilities and their surroundings, in the morning and afternoon shifts.

Room	Variables	Morning ^‡^				Afternoon ^‡^			
		n (%)	Not Adequate	Adequate	*p*	n (%)	Not Adequate	Adequate	*p*
			n (%)	n (%)			n (%)	n (%)	
Reception Room	Total	23 (100)	14 (60.9)	9 (39.1)		23 (100)	16 (69.6)	7 (30.4)	
	Lightweight partition wall	4 (17.4)	3 (21.4)	1 (11.1)	1.000 ^a^	4 (17.4)	3 (18.8)	1 (14.3)	1.000 ^a^
	Concrete slab	18 (78.3)	12 (85.7)	6 (66.7)	0.343 ^a^	18 (78.3)	12 (75.0)	6 (85.7)	1.000 ^a^
	Lightweight ceiling panels	5 (21.7)	2 (14.3)	3 (33.3)	0.343 ^a^	5 (21.7)	4 (25.0)	1 (14.3)	1.000 ^a^
	Front door open	19 (82.6)	10 (71.4)	9 (100)	0.127 ^a^	22 (95.7)	15 (93.8)	7 (100)	1.000 ^a^
	Outside window open	14 (60.9)	8 (57.1)	6 (66.7)	1.000 ^a^	11 (47.8)	7 (43.8)	4 (57.1)	0.667 ^a^
	Internal curtains open	3 (13.0)	0 (0.0)	3 (33.3)	0.047 ^a^	3 (13.0)	2 (12.5)	1 (14.3)	1.000 ^a^
	Ceiling light on	20 (87.0)	11 (78.6)	9 (100)	0.253 ^a^	20 (87.0)	13 (81.3)	7 (100)	0.526 ^a^
	Sunny weather	10 (43.5)	8 (57.1)	2 (22.2)	0.197 ^a^	10 (43.5)	8 (50.0)	2 (28.6)	0.405 ^a^
	Cloudy weather	8 (34.8)	3 (21.4)	5 (55.6)	0.179 ^a^	6 (26.1)	4 (25.0)	2 (28.6)	1.000 ^a^
Vaccination Room	Total	23 (100)	14 (60.9)	9 (39.1)		23 (100)	11 (47.8)	12 (52.2)	
	Lightweight partition wall	1 (4.3)	1 (7.1)	0 (0.0)	1.000 ^a^	1 (4.3)	1 (9.1)	0 (0.0)	0.478 ^a^
	Concrete slab	18 (78.3)	10 (71.4)	8 (88.9)	0.611 ^a^	18 (78.3)	9 (81.8)	9 (75.0)	1.000 ^a^
	Lightweight ceiling panels	5 (21.7)	4 (28.6)	1 (11.1)	0.611 ^a^	5 (21.7)	2 (18.2)	3 (25.0)	1.000 ^a^
	Front door open	18 (78.3)	10 (71.4)	8 (88.9)	0.611 ^a^	19 (82.6)	9 (81.8)	10 (83.3)	1.000 ^a^
	Outside window open	9 (39.1)	6 (42.9)	3 (33.3)	1.000 ^a^	8 (34.8)	4 (36.4)	4 (33.3)	1.000 ^a^
	Indoor curtain open	8 (34.8)	6 (42.9)	2 (22.2)	0.400 ^a^	9 (39.1)	4 (36.4)	5 (41.7)	1.000 ^a^
	Ceiling light on	22 (95.7)	13 (92.9)	9 (100)	1.000 ^a^	21 (91.3)	10 (90.9)	11 (91.7)	1.000 ^a^
	Sunny weather	10 (43.5)	5 (35.7)	5 (55.6)	0.417 ^b^	11 (47.8)	4 (36.4)	7 (58.3)	0.525 ^b^
	Cloudy weather	7 (30.4)	5 (35.7)	2 (22.2)	0.657 ^a^	5 (21.7)	4 (36.4)	2 (8.3)	0.155 ^a^
Consultation room	Total	23 (100)	19 (82.6)	4 (17.4)		23 (100)	18 (78.3)	5 (21.7)	
	Lightweight partition wall	3 (13.0)	3 (15.8)	0 (0.0)	1.000 ^a^	3 (13.0)	3 (16.7)	0 (0.0)	1.000 ^a^
	Concrete slab	15 (65.2)	12 (63.2)	3 (75.0)	1.000 ^a^	15 (65.2)	11 (61.1)	4 (80.0)	0.621 ^a^
	Lightweight ceiling panels	8 (34.8)	7 (36.8)	1 (25.0)	1.000 ^a^	8 (34.8)	7 (38.9)	1 (20.0)	0.621 ^a^
	Front door open	18 (78.3)	15 (78.9)	3 (75.0)	1.000 ^a^	18 (78.3)	14 (77.8)	4 (80.0)	1.000 ^a^
	Outside window open	6 (26.1)	5 (26.3)	1 (25.0)	1.000 ^a^	7 (30.4)	7 (38.9)	0 (0.0)	0.272 ^a^
	Indoor curtain open	4 (17.4)	3 (15.8)	1 (25.0)	1.000 ^a^	5 (21.7)	3 (16.7)	2 (40.0)	0.291 ^a^
	Ceiling light on	19 (82.6)	16 (84.2)	3 (75.0)	1.000 ^a^	12 (52.2)	8 (44.4)	4 (80.0)	0.317 ^a^
	Sunny weather	10 (43.5)	8 (42.1)	2 (50.0)	1.000 ^a^	11 (47.8)	10 (55.6)	1 (20.0)	0.317 ^a^
	Cloudy weather	6 (26.1)	4 (21.1)	2 (50.0)	0.270 ^a^	6 (26.1)	3 (16.7)	3 (60.0)	0.089 ^a^
Procedure room	Total	23 (100)	21 (91.3)	2 (8.7)		23 (100)	20 (87.0)	3 (13.0)	
	Lightweight partition wall	3 (13.0)	3 (14.3)	0 (0.0)	1.000 ^a^	3 (13.0)	2 (10.0)	1 (33.3)	0.356 ^a^
	Concrete slab	17 (73.9)	16 (76.2)	1 (50.0)	0.462 ^a^	17 (73.9)	16 (80.0)	1 (33.3)	0.155 ^a^
	Lightweight ceiling panels	6 (26.1)	5 (23.8)	1 (50.0)	0.462 ^a^	6 (26.1)	4 (20.0)	2 (66.7)	0.155 ^a^
	Front door open	21 (91.3)	19 (90.5)	2 (100)	1.000 ^a^	19 (82.6)	17 (85.0)	2 (66.7)	0.453 ^a^
	Outside window open	11 (47.8)	10 (47.6)	1 (50.0)	1.000 ^a^	12 (52.2)	10 (50.0)	2 (66.7)	1.000 ^a^
	Indoor curtain open	5 (21.7)	5 (23.8)	0 (0.0)	1.000 ^a^	5 (21.7)	5 (25.0)	0 (0.0)	1.000 ^a^
	Ceiling light on	17 (73.9)	15 (71.4)	2 (100)	1.000 ^a^	17 (73.9)	14 (70.0)	3 (100)	0.539 ^a^
	Sunny weather	9 (39.1)	7 (33.3)	2 (100)	0.142 ^a^	9 (39.1)	7 (35.0)	2 (66.7)	0.538 ^a^
	Cloudy weather	9 (39.1)	9 (42.9)	0 (0.0)	0.502 ^a^	6 (26.1)	5 (25.0)	1 (33.3)	1.000 ^a^

^a^ Fisher’s Exact test; ^b^ Chi-square’s test; ^‡^ Zone Time: UTC-3 BRT (Brasília Time); Adequate = reception room, lighting levels of 200 Lux or 300 Lux, and for other rooms, lighting levels of 300 Lux or 500 Light. Not adequate = values above or below those indicated.

**Table 5 ijerph-21-00847-t005:** Results concerning indoor temperature, relative humidity, noise, and illuminance in both shifts of 23 PHC facilities.

Room	Variable	* Assessment in 2 Shifts ^‡^		
		No Adequate	Partially Adequate	Fully Adequate
		n (%)	n (%)	n (%)
Reception room	Temperature/Humidity	19 (82.6)	4 (17.4)	0 (0.0)
	Noise	21 (91.3)	1 (4.3)	1 (4.3)
	Lighting	11 (47.8)	8 (34.8)	4 (17.4)
Vaccination room	Temperature/Humidity	18 (78.3)	4 (17.4)	1 (4.3)
	Noise	19 (82.6)	4 (17.4)	0 (0.0)
	Lighting	9 (39.1)	7 (30.4)	7 (30.4)
Consultation room	Temperature/Humidity	19 (82.6)	4 (17.4)	0 (0.0)
	Noise	22 (95.7)	0 (0.0)	1 (4.3)
	Lighting	15 (65.2)	7 (30.4)	1 (4.3)
Procedure room	Temperature/Humidity	20 (87.0)	1 (4.3)	2 (8.7)
	Noise	22 (95.7)	1 (4.3)	0 (0.0)
	Lighting	18 (78.3)	5 (21.7)	0 (0.0)

* Fully adequate = 2 shifts; Partially adequate = one shift; and Not adequate = both shifts; ^‡^ Zone Time: UTC-3 BRT (Brasília Time).

**Table 6 ijerph-21-00847-t006:** Associations between the perceptions of health professionals working in PHC facilities and the adequacy of physical parameters measured in the same PHC facilities.

Variables Perceived by Health Professionals	Physical Parameters Measured in the PHC Facilities (Percentage of Adequacy)			
	Temperature/ Moisture	Noise	Lighting	General
Percentages perceived regarding the physical structure of the PHC facility				
Barrier of any intensity	r_s_ = 0.080; *p* = 0.717	r_s_ = −0.135; *p* = 0.538	r_s_ = −0.093; *p* = 0.673	r_s_ = −0.244; *p* = 0.261
Facilitator of any intensity	r_s_ = −0.093; *p* = 0.672	r_s_ = 0.172; *p* = 0.432	r_s_ = −0.158; *p* = 0.471	r_s_ = 0.070; *p* = 0.751
Percentage of perceived physical risk of any intensity at PHC facility	r_s_ = 0.284; *p* = 0.189	r_s_ = 0.049; *p* = 0.823	r_s_ = −0.296; *p* = 0.170	r_s_ = −0.073; *p* = 0.739
Percentage of physical risks perceived at PHC facility				
Uncomfortable room temperature	r_s_ = 0.302; *p* = 0.162			r_s_ = 0.224; *p* = 0.304
Annoying and irritating noise		r_s_ = 0.067; *p* = 0.761		r_s_ = 0.066; *p* = 0.765
Poor lighting (illuminance)			r_s_ = −0.121; *p* = 0.581	r_s_ = −0.302; *p* = 0.161
Lack of ventilation in the environments	r_s_ = 0.123; *p* = 0.576			r_s_ = −0.163; *p* = 0.457

r_s_ = Spearman’s correlation coefficient.

**Table 7 ijerph-21-00847-t007:** Associations between the sociodemographic and labor data of health professionals who work at the PHC facility and physical parameters measured of the same PHC facility.

Sociodemographic and Labor Data	Physical Parameters Measured in the PHC Facilities (Percentage of Adequacy)			
	Temperature/ Moisture	Noise	Lighting	General
Average age (years)	r_s_ = −0.110; *p* = 0.618	r_s_ = −0.112; *p* = 0.610	r_s_ = −0.103; *p* = 0.639	r_s_ = −0.057; *p* = 0.796
Percentage of female	r_s_ = 0.091; *p* = 0.681	r_s_ = 0.139; *p* = 0.527	r_s_ = −0.038; *p* = 0.865	r_s_ = 0.195; *p* = 0.373
Percentage of nurses	r_s_ = 0.227; *p* = 0.298	r_s_ = 0.250; p = 0.249	r_s_ = 0.346; *p* = 0.106	r_s_ = 0.634; *p* = 0.001
Percentage of physicians	r_s_ =−0.314; *p* = 0.144	r_s_ = −0.089; *p* = 0.686	r_s_ = −0.067; *p* = 0.762	r_s_ = −0.408; *p* = 0.053
Percentage of nursing technician/assistant	r_s_ =−0.233; *p* = 0.285	r_s_ = 0.003; *p* = 0.988	r_s_ = −0.267; *p* = 0.219	r_s_ = −0.278; *p* = 0.200
Percentage of community health agent	r_s_ = 0.083; *p* = 0.707	r_s_ = −0.268; *p* = 0.217	r_s_ = −0.063; *p* = 0.776	r_s_ = −0.137; *p* = 0.534
Percentage of dentist	r_s_ = 0.194; *p* = 0.375	r_s_ = 0.227; *p* = 0.297	r_s_ = 0.197; *p* = 0.367	r_s_ = 0.213; *p* = 0.329
Percentage of oral health technician/assistant	r_s_ = 0.180; *p* = 0.411	r_s_ = 0.015; *p* = 0.945	r_s_ = 0.294; *p* = 0.273	r_s_ = 0.367; *p* = 0.085
Average time working in the profession (years)	r_s_ = 0.143; *p* = 0.514	r_s_ = 0.195; *p* = 0.373	r_s_ = −0.071; *p* = 0.748	r_s_ = 0.235; *p* = 0.280
Average time working at ABS (years)	r_s_ = 0.129; *p* = 0.558	r_s_ = 0.142; *p* = 0.518	r_s_ = −0.113; *p* = 0.608	r_s_ = 0.147; *p* = 0.504
Percentage of daytime working hours at ABS	r_s_ = 0.243; *p* = 0.264	r_s_ = 0.217; *p* = 0.320	r_s_ = −0.026; *p* = 0.906	r_s_ = 0.250; *p* = 0.249
Percentage of other work parallel to ABS	r_s_ = −0.310; *p* = 0.149	r_s_ = 0.061; *p* = 0.782	r_s_ = 0.191; *p* = 0.384	r_s_ = 0.042; *p* = 0.850
Percentage with up to secondary education	r_s_ = −0.054; *p* = 0.806	r_s_ = 0.083; *p* = 0.705	r_s_ = −0.043; *p* = 0.847	r_s_ = 0.145; *p* = 0.509
Percentage with incomplete/complete higher education/technology degree	r_s_ = 0.065; *p* = 0.769	r_s_ = −0.007; *p* = 0.974	r_s_ = −0.112; *p* = 0.612	r_s_ = −0.270; *p* = 0.212
Percentage with specialization/Master’s/ Doctorate degree	r_s_ = 0.045; *p* = 0.838	r _s_ = −0.005; *p* = 0.984	r_s_ = 0.391; *p* = 0.065	r_s_ = 0.354; *p* = 0.098

r_s_ = Spearman correlation coefficient.

**Table 8 ijerph-21-00847-t008:** Multivariate Linear Regression Analysis to evaluate factors independently associated with the percentage of adequacy of the physical parameters of the PHC facilities analyzed.

Variables	b (95% CI)	Beta	*p*	R^2^
Temperature/humidity adequacy percentage in the PHC facilities				26.6%
Percentage of perceived physical risk of any intensity	0.34 (−0.82 to 1.51)	0.128	0.542	
Uncomfortable room temperature percentage	0.41 (−0.12 to 0.94)	0.335	0.120	
Percentage of physicians	−0.95 (−1.88 to −0.03)	−0.444	0.044	
Percentage of other work parallel to PHC	−0.38 (−1.33 to 0.57)	−0.195	0.411	
Lighting adequacy percentage in the PHC facilities				19.0%
Percentage of perceived physical risk of any intensity	−0.27 (−1.18 to 0.64)	−0.140	0.540	
Percentage of nurses in the PHC facilities	0.33 (−0.58 to 1.24)	0.181	0.453	
Percentage with Specialization/Master’s/Doctorate degree	0.58 (−0.05 to 1.21)	0.386	0.069	
General adequacy percentage in the PHC facilities				60.3%
Percentage of poor lighting perceived in the PHC facilities	0.08 (−0.11 to 0.26)	0.162	0.389	
Percentage of nurses	0.69 (0.25 to 1.13)	0.605	0.004	
Percentage of doctors	−0.33 (−0.68 to 0.02)	−0.342	0.066	
Percentage of oral health technician/assistant	0.74 (0.08 to 1.40)	0.389	0.031	
Percentage with specialization/Master’s/Doctorate degree	−0.03 (−0.40 to 0.33)	−0.036	0.847	

b = regression coefficient; 95% CI = 95% confidence interval; Beta = standardized regression coefficient; R^2^ = coefficient of determination.

## Data Availability

Data regarding this study can be provided upon request to the corresponding author. Data are not publicly available due to ethical issues.

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
