# Peer review of "The Working Environment in Primary Healthcare Outpatient Facilities: Assessment of Physical Factors and Health Professionals’ Perceptions of Working Environment Conditions"

_ijerph, 2024, doi:10.3390/ijerph21070847_

Round 1

Reviewer 1 Report (Previous Reviewer 2)

Comments and Suggestions for Authors

I don't have any supplementary comments

Author Response

Dear Reviewer,

We hope this email finds you well.

Thank you for your valuable feedback on our manuscript (ijerph-2777202). We are grateful for your work time. Your input has been instrumental in refining our work, and we are delighted to have incorporated your recommendations. 

Best wishes,
Corresponding Author and Co-Author

Reviewer 2 Report (Previous Reviewer 1)

Comments and Suggestions for Authors

Nothing to add to previous review

Comments on the Quality of English Language

Nothing to add to previous review

Author Response

Dear Reviewer,
We hope this email finds you well. We want to inform you that we have made changes and additions to the Introduction and Materials and Methods sections. All changes were highlighted in red letters. We hope we have fulfilled your previous recommendations. Your valuable input has been instrumental in refining our work; we appreciate your time working.
Stay well.
Best regards,
Corresponding Author and (Co-Authors)

Reviewer 3 Report (New Reviewer)

Comments and Suggestions for Authors

1. How would you characterize the objectives of the study?  1) adequacy of physical factors, 2) assessment of the association between environmental conditions, 3) analysis of professionals' perceptions of physical exposure, 4) estimation of the association between environmental conditions in PHC facilities.
2. I recommend that you express the purpose of your research clearly and concisely. Given the large number of variables we are dealing with, such an expression is likely to be important as a device to aid the reader's understanding. 
3. Please consider separating the study design, study population, and setting on line 141 into separate headings; this would be a strategy to make each more clear. 
4. You may want to reconsider the organization of the Methods section. 
It is necessary to consistently arrange the research items as sub-items of '2. measure'.
'2.2.2 Measurement Criteria for Analysis' should be renamed to 'Data Analysis'. It is recommended to list the criteria for analysis afterwards (1) Measurement Criteria for Analysis / (2) Statistical Methods.   The list of sub-numbers in this section should be revised to be consistent. 

5. The statements on lines 407 through 411 fall under "Ethical considerations of the study. Please provide a separate subheading. 
6. Please indicate the time zone for morning and afternoon in Table 1(including table 3). 
7. In the variables section of Table 1, total and its sub-variables (details) seem to need to be separated. Please consider indentation, etc. 
8. Show what the horizontal axis in Figure 2 means. At the bottom of the horizontal axis, at the "PRIMARY HEALTH CARE FACILITY" location, it seems like there should be a heading that explains what the horizontal axis means.

9. Paragraphs 712-714 make me wonder where the focus of this study is. If the focus is on the appropriateness of the work environment, I think this is outside the scope of the study's interpretation. If you want to talk about the impact of the adequacy of the work environment on humans, the scope of the study seems to be very broad.   

Author Response

Dear Reviewer,

We are sincerely grateful for your expert and thoughtful feedback on our manuscript (ijerph-2777202). Your valuable input has been instrumental in refining our work, and we are delighted to have incorporated your recommendations.

Wishing you continued success and well-being.

Corresponding Author and (Co-Authors)

The comments point by point:

  • Reviewer – “1.How would you characterize the objectives of the study? 1) adequacy of physical factors, 2) assessment of the association between environmental conditions, 3) analysis of professionals' perceptions of physical exposure, 4) estimation of the association between environmental conditions in PHC facilities.”

Authors’ response: Thank you for your recommendation. We have revised the wording of the study's objectives. We took care not to alter our original purposes and have added an additional objective: a) to assess the adequacy of physical parameters/factors (temperature, relative humidity, noise, and illuminance levels) of the work environment in facilities PHC; b) to evaluate the association between the adequacy of these measured physical parameters and the physical characteristics of the PHC facilities and their surroundings; c) to evaluate the association between health professionals' perceptions about exposure to physical risks in the PHC work environment and the adequacy of physical parameters measured in the same facilities. (p. 3)

  • Reviewer – “2.I recommend that you express the purpose of your research clearly and concisely. Given the large number of variables we are dealing with, such an expression is likely to be important as a device to aid the reader's understanding.”

Authors’ response: Thank you for your recommendation. We believe the three objectives still address the main question of the study: “…it was possible to question whether the PHC facilities with the highest percentage of adequacy regarding the four criteria (for parameters) were also those with the lowest percentage of health professionals who perceived physical risk in the work environment of the PHC facilities” (p.3). Additionally, we added three secondary hypotheses to detail the relationships between the various variables included in the analyses. We hope these additions and changes will help clarify the primary purpose of the study and its supplementary components.

  • Reviewer – “3. Please consider separating the study design, study population, and setting on line 141 into separate headings; this would be a strategy to make each more clear.”

Authors’ response: Thank you for your recommendation. We have included an explanatory subsection in the Materials and Methods section to describe the origin of the current study (2.1. Origin of the current study). We provided detailed information about the study, including the population, sample, and location. We chose this approach to address the concerns of both reviewers. Please evaluate if our method meets your criteria. Thank you very much.

  • Reviewer – “4. You may want to reconsider the organization of the Methods section. It is necessary to consistently arrange the research items as sub-items of '2. measure'. '2.2.2 Measurement Criteria for Analysis' should be renamed to 'Data Analysis'. It is recommended to list the criteria for analysis afterwards (1) Measurement Criteria for Analysis / (2) Statistical Methods. The list of sub-numbers in this section should be revised to be consistent.”

Authors’ response: Thank you for your recommendation. We have made all the requested changes to the items and their contents. Additionally, we have added further details regarding the study's methodological process. We hope we have met your expectations. Please review our revisions, which can be found in the Materials and Methods section on pages 4 to 9.

  • Reviewer – “5. The statements on lines 407 through 411 fall under "Ethical considerations of the study. Please provide a separate subheading.”

Authors’ response: Thank you for your recommendation. We have implemented the requested changes. (2.5 Ethical considerations of the study) p.9.

  • Reviewer – “6. Please indicate the time zone for morning and afternoon in Table 1(including table 3).”

Authors’ response: We appreciate your recommendation. We have included the time zone (‡Zone Time: UTC -3 BRT (Brasília Time)) in all tables where the morning and afternoon shift variables were used.

  • Reviewer – “7. In the variables section of Table 1, total and its sub-variables (details) seem to need to be separated. Please consider indentation, etc.”

Authors’ response: Thank you for your recommendation. We have made the necessary revisions to Table 1, which can be reviewed on page 11.

  • Reviewer – “8. Show what the horizontal axis in Figure 2 means. At the bottom of the horizontal axis, at the "PRIMARY HEALTH CARE FACILITY" location, it seems like there should be a heading that explains what the horizontal axis means.”

Authors’ response: Thank you for your recommendation. As suggested, we have provided an additional explanation in Figure 2.

  • Reviewer – “Paragraphs 712-714 make me wonder where the focus of this study is. If the focus is on the appropriateness of the work environment, I think this is outside the scope of the study's interpretation. If you want to talk about the impact of the adequacy of the work environment on humans, the scope of the study seems to be very broad.”

Authors’ response: Thank you for your comment. We have reassessed the ideas expressed in the paragraph (between lines 712-714) mentioned by the reviewer and have removed them from the manuscript due to the study's own limitations. The findings of the study underscore the significance of research in this area. While there are studies conducted in various work settings documented in the literature, there remains a scarcity of research specifically focusing on PHC environments. Furthermore, our study's results and conclusions can be compared with those of other national and international studies.

We reiterate our agreement for your work time.

Reviewer 4 Report (New Reviewer)

Comments and Suggestions for Authors

The topic is okay; the abstract and introduction are acceptable. The significance level was 95% confidence and 5% errors (p ≤ 0.05) (it is the same as the significance was set at 5%; p ≤ 0.05). How did the researchers adopt 210 health professionals working in these facilities in southern Brazil, online questionnaires? Definite or indefinite population, sample size determination, and sampling technique are necessary to identify. The first sampling is to select Rio Grande do Sul, Brazil, by purposive sampling, Please identify the criteria for choosing this city. The second step is to identify the sample size determination and sampling technique: probability or nonprobability? How did the researchers test the validity of survey questions, and which studies were used for question development? Which statistical analysis is used for each of tables 1, 2 and 3? Please add and identify each table. Please avoid using we, but adopt researchers instead throughout the paper. The survey questions are not shown, it seems like the Likert Rating Scale has not been adopted, but use the questions and evaluation in percentage to test the hypotheses. Please ensure that discussions and conclusions include variables and hypotheses (because the results show a p-value, the hypotheses could be identified). The references are acceptable.

Author Response

Dear Reviewer,

Thank you for your valuable feedback on our manuscript (ijerph-2777202). Your input has been instrumental in refining our work, and we are delighted to have incorporated your recommendations.

Best wishes,

Corresponding Author and Co-Authors

The comments point by point:

(1) Reviewer – “The topic is okay; the abstract and introduction are acceptable.”

Authors’ response: Thank you for your comment. We would like to inform you that, in response to the reviewers' suggestions, we have made changes to the abstract and introduction. Please review them in this revision. Thank you.

(2)       Reviewer – “The significance level was 95% confidence and 5% errors (p ≤ 0.05) (it is the same as the significance was set at 5%; p ≤ 0.05). How did the researchers adopt 210 health professionals working in these facilities in southern Brazil, online questionnaires? Definite or indefinite population, sample size determination, and sampling technique are necessary to identify. The first sampling is to select Rio Grande do Sul, Brazil, by purposive sampling, Please identify the criteria for choosing this city. The second step is to identify the sample size determination and sampling technique: probability or nonprobability?”

Authors’ response: We appreciate your key points and recommendations regarding aspects of the Materials and Methods section. We want to inform you that we have revised the methodological procedures, refining the way they were explained, and included additional details throughout this section. Specifically, we have emphasized specific aspects concerning the population, definition of the non-probabilistic sampling, sample selection through consecutive intentional sampling and its inclusion and exclusion criteria, selection of the study site, recruitment of professionals interviewed face-to-face at their workplaces, as well as detailed information about the PHC units monitored in their physical parameters. The additions and changes have been highlighted in red font. The alterations in the Materials and Methods section can be found between pages 4-9 of the manuscript. Furthermore, we would like to inform you that the modifications and additions made have addressed the feedback from both reviewers.

(3) Reviewer – “How did the researchers test the validity of survey questions, and which studies were used for question development?”

Authors’ response: We appreciate your inquiries. We have made it clearer that the questionnaire used in the broader study has been utilized in other research studies by the research group, and a pilot study was conducted. The form for collecting data on the physical parameters of the PHC unit was developed by a multidisciplinary team consisting of research professors with expertise in the fields of engineering, indoor environmental comfort, and public health. We utilized a framework of national and international technical and scientific references. It's worth emphasizing that throughout the described methodological process, we included references that supported the theoretical and operational concepts of the study. For the cross-sectional study regarding the monitored physical parameters in the PHC units, we also conducted a pilot study. However, our instrument has not yet been translated into English.

(4) Reviewer - “Which statistical analysis is used for each of tables 1, 2 and 3? Please add and identify each table.”

Authors’ response: Thank you for your recommendation and attention. We have reviewed all tables and included the performed statistical analysis.

(5) Reviewer – “Please avoid using we, but adopt researchers instead throughout the paper.”

Authors’ response: Thank you for your attention and request. We have made the modifications.

(6) Reviewer – “The survey questions are not shown, it seems like the Likert Rating Scale has not been adopted, but use the questions and evaluation in percentage to test the hypotheses.”

Authors’ response: Thank you for your comment and recommendation. We have clarified in the manuscript text the relationship between the main question as the study's purpose and have included secondary hypotheses in the study. “This current study is composed of two integrated parts: the first is a cross-sectional study that measures the physical parameters/factors (temperature, relative humidity, noise, and illuminance levels) of the working environment of PHC facilities, and the second is an ecological survey that analyzes the perception of health professionals regarding exposure to physical risk factors of the work environment in the same PHC facilities studied.” (p.4). Based on this explanation, we constructed the analyses by integrating the two sets of data and formulated the objectives and secondary hypotheses around the main question. Thus, we understand the study's main question “…it was possible to question whether the PHC facilities with the highest percentage of adequacy regarding the four criteria (for parameters) were also those with the lowest percentage of health professionals who perceived physical risk in the work environment of the PHC facilities” (p.3). Additionally, we added three secondary hypotheses as a mechanism to detail the relationships between the various variables included in the analyses. We hope that these additions and changes will allow for a better understanding of the main purpose of the study and its complements.

(7) Reviewer – “Please ensure that discussions and conclusions include variables and hypotheses (because the results show a p-value, the hypotheses could be identified).”

Authors’ response: Thank you for your recommendations. We want to inform you that we have made changes and additions to the discussion and conclusion sections to ensure they are aligned with the secondary hypotheses and our central question. We are pleased with the connections between the results and interpretations concluded by the study, as reiterated in the text.

(8) Reviewer – “The references are acceptable.”

Authors’ response: We appreciate your comments and suggestions regarding the structure of our study. The evidence from our study highlights the importance of investigations in PHC facilities environments, an area that has yet to be explored in the literature regarding the topic addressed. We consider studies carried out in other work environments, but our focus remains PHC. Furthermore, the results and conclusions of our research can provide a valuable point of comparison with different national and international studies.

We are grateful for your work time in reviewing our work.

Round 2

Reviewer 4 Report (New Reviewer)

Comments and Suggestions for Authors

Acceptable.

Comments on the Quality of English Language

Minor changes are required.

Author Response

Dear Reviewer,
Thank you for your valuable feedback on our manuscript (ijerph-2963485). Your input has been essential in refining our work, and we are satisfied to have incorporated your recommendations.
Best regards,
Corresponding Author and Co-Authors

This manuscript is a resubmission of an earlier submission. The following is a list of the peer review reports and author responses from that submission.

Round 1

Reviewer 1 Report

Comments and Suggestions for Authors

The authors are to be commended for a painstaking report on the physical parameters affecting the comfort, safety and productivity of workers in Primary Health care facilities in a city in southern Brazil.  They have also reported on the perception of staff of their level of comfort and of safety; the latter data were collected prior to the measurement of physical factors, and it is unclear whether this information was obtained in similar climatic conditions. The thermal conditions were measured in the fall, at which time it appears the climate was mild – around 180C – while it is not known whether the staff were interviewed at a similar time, or instead at a time of much warmer weather.

One of the two major aims of this study is to determine whether the level of exposure to heat, humidity, noise etc is compliant with relevant regulatory or recommended standards, and this is reported on in a high level of detail. While these findings are likely to be of importance to those responsible for the health and welfare of PHC staff in that particular study, they do not draw any scientific inferences which are of general application.  Articles to scientific journals should present conclusions which are applicable generally, and not just to the area and population under study.  This study is of importance to the local occupational health authorities, but does not contain information of interest even to PHC staff elsewhere, let alone to other workplaces.

In summary, this will be a valuable report for those concerned with the comfort, safety and productivity of the workforce under study, but lacks the generalisability of findings required for an international scientific journal.

Comments on the Quality of English Language

The authors are to be commended for a painstaking report on the physical parameters affecting the comfort, safety and productivity of workers in Primary Health care facilities in a city in southern Brazil.  They have also reported on the perception of staff of their level of comfort and of safety; the latter data were collected prior to the measurement of physical factors, and it is unclear whether this information was obtained in similar climatic conditions. The thermal conditions were measured in the fall, at which time it appears the climate was mild – around 180C – while it is not known whether the staff were interviewed at a similar time, or instead at a time of much warmer weather.

One of the two major aims of this study is to determine whether the level of exposure to heat, humidity, noise etc is compliant with relevant regulatory or recommended standards, and this is reported on in a high level of detail. While these findings are likely to be of importance to those responsible for the health and welfare of PHC staff in that particular study, they do not draw any scientific inferences which are of general application.  Articles to scientific journals should present conclusions which are applicable generally, and not just to the area and population under study.  This study is of importance to the local occupational health authorities, but does not contain information of interest even to PHC staff elsewhere, let alone to other workplaces.

In summary, this will be a valuable report for those concerned with the comfort, safety and productivity of the workforce under study, but lacks the generalisability of findings required for an international scientific journal.

Reviewer 2 Report

Comments and Suggestions for Authors

In the paper titled "The physical factors of working environments in Primary Health Care facilities and health workers’ perceptions of whether these factors represent barriers or facilitators to the work performed in these facilities" the authors aim to establish a connection between objective factors and subjective perceptions related to the work environment. The manuscript is highly detailed in its description but takes on the appearance of a report rather than a scientific article. Nevertheless, the authors' effort and substantial work are evident.

I do not believe that the current structure of the paper would be engaging for the reader, as it lacks a crucial link between what is objectively measured and what is perceived by the survey respondents. Alternatively, the work could be divided into two parts: one part describing the measurement of exposure (as already outlined) and a second part considering the survey results and their correlation with the measurements.

I would like to commend the authors once again for their significant efforts and the overall project.